# Arabidopsis conditional photosynthesis mutants *abc1k1* and *var2* accumulate partially processed thylakoid preproteins and are defective in chloroplast biogenesis
Joy Collombat[1], Manfredo Quadroni [2], Véronique Douet[1], Rosa Pipitone[3], Fiamma Longoni[1] & Felix Kessler [1] ✉

Photosynthetic activity is established during chloroplast biogenesis. In this study we used 680 nm red light to overexcite Photosystem II and disrupt photosynthesis in two conditional mutants (*var2* and *abc1k1*) which reversibly arrested chloroplast biogenesis. During biogenesis, chloroplasts import most proteins associated with photosynthesis. Some of these must be inserted in or transported across the thylakoid membrane into the thylakoid lumen. They are synthesized in the cytoplasm with cleavable targeting sequences and the lumenal ones have bi-partite targeting sequences (first for the chloroplast envelope, second for the thylakoid membrane). Cleavage of these peptides is required to establish photosynthesis and a critical step of chloroplast biogenesis. We employ a combination of Western blotting and mass spectrometry to analyze proteins in *var2* and *abc1k1*. Under red light, *var2* and *abc1k1* accumulated incompletely cleaved processing intermediates of thylakoid proteins. These findings correlated with colorless cotyledons, and defects in both chloroplast morphology and photosynthesis. Together the results provide evidence for the requirement of active photosynthesis for processing of photosystem-associated thylakoid proteins and concomitantly progression of chloroplast biogenesis.

Plant embryos inside seeds contain undifferentiated, non-photosynthetic proplastids. During the transition to photosynthetically active chloroplasts ("chloroplast biogenesis")[1] the organelle needs to import hundreds of different nuclear-encoded proteins from the cytosol[2,3]. N-terminal transit sequences enable their translocation across the dual membrane envelope by the TOC-TIC (Translocon of the Outer and Inner membranes of the Chloroplasts) system[4–6]. Proteins that function in the photosynthetic light reactions must further be inserted in or translocated across the thylakoid membrane and assembled into the photosynthetic apparatus consisting of Photosystem II (PSII) associated with the oxygen-evolving complex (OEC), Photosystem I (PSI), cytochrome b$_6$f complex and the ATPase complex and associated antenna complexes. This involves three transport systems, the signal recognition particle (SRP), SEC (Secretory), and twin arginine

translocation (TAT) pathways, that are conserved from cyanobacteria[3,7–10]. Insertion into the thylakoid membrane is mediated by the cp-SRP pathway[11,12]. Translocation across the thylakoid membrane and into the thylakoid lumen depends on either the SEC or TAT pathways[10,13,14]. This concerns around 80 proteins in *Arabidopsis thaliana*[15–19]. Lumenal thylakoid proteins carry a bipartite N-terminal targeting peptide[20]. The first, most N-terminal part is a transit peptide, responsible for the import into the chloroplast. It is cleaved by stromal processing peptidase (SPP)[21–23]. The second part, exposed upon cleavage of the transit peptide, enables transport across the thylakoid membrane into the lumen mediated by the SEC or TAT pathways (the TAT signal contains a twin-arginine "RR" but otherwise resembles that of SEC)[24–26]. These peptides are cleaved by a type I signal peptidase known as thylakoid processing peptidase (TPP)[27–29]. Only one of

[1]Institute of biology, Plant Physiology Laboratory, Université de Neuchâtel, 2000 Neuchâtel, Switzerland. [2]Protein Analysis Facility (PAF), Université de Lausanne, 1015 Lausanne, Switzerland. [3]Thermo Fisher, 39 rue d'Armagnac, 33800 Bordeaux, France. ✉e-mail: felix.kessler@unine.ch

three TPP homologs, plastidic type I signal peptidase 1 (Plsp1) is essential[30]. The *plsp1* mutant is seedling lethal and causes strong thylakoid development defects demonstrating the importance of lumenal protein processing for chloroplast biogenesis[30]. Plsp1 is known to be involved in the processing of the TOC75 protein (Translocon of the Outer membranes of the Chloroplasts of 75 kDa) and three lumenal proteins: plastocyanin, OE33 (PsbO1), and OE23 (PsbP)[30,31]. The oxygen-evolving complex components OE33 and OE23 are amongst the most abundant chloroplast proteins[16,19,32]. In vascular plants, the OEC is composed of three proteins for which two homologs each exist: PsbO-1/-2 (OEC33), PsbP-1/-2 (OEC23), and PsbQ-1/-2 (OEC16). They are peripheral membrane proteins facing the thylakoid lumen and are associated with the D1 (PsbA) protein at the core of PSII[33,34]. PsbO (OEC33) utilizes the SEC pathway whereas PsbP and PsbQ require the TAT pathway[29,35]. Components of the plastocyanin-docking site of PSI, PsaF, and PsaN, are also peripheral membrane proteins facing the thylakoid lumen and requiring the SEC and TAT pathways respectively[36,37]. The energy required by SEC-dependent translocation is provided by ATP[38,39] and that of the TAT pathway (also known as ∂pH pathway) by the proton gradient across the thylakoid membrane[29,40,41] both resulting primarily from photosynthesis. Translocation, processing, and assembly of photosynthesis-associated proteins such as the OEC components is crucial for chloroplast biogenesis[30]. But once assembled the photosynthetic machinery must also be maintained. The PsbA (D1) protein can easily be damaged by charge recombination and must then be replaced under light stress conditions[42,43]. The importance of this process is evident in the variegated (white and green leaf sectors) phenotype of *var2 (ftsh2)* mutant that lacks the FTSH2 protease required for the turnover of D1 and the repair of PSII. However, reduced levels of the thylakoid FTSH complex remain as the other subunits FTSH1, FTSH5, and FTSH8 are still present and prevent the lethal phenotype observed in the complete absence of the FTSH complex[44,45].

Similar to *var2*, the *abc1k1* mutant, first identified as *pgr6* (proton regulation 6), has a conditional variegated phenotype under high light[46]. *abc1k1* is defective for NPQ ("non-photochemical quenching" of chlorophyll) and shows impaired photosynthetic electron transport[46–48]. Interestingly, the *abc1k1* mutant was also identified as *bdr1-1* (Bleached Dwarf under red light 1–1) in a genetic screen for mutants affected in development under red light and had a colorless phenotype[49,50]. ABC1K1 is an atypical kinase associated with plastoglobules (PG), lipid droplets associated with the thylakoid membrane. While the conserved active site aspartate residue is needed[51], the knowledge on ABC1K1 kinase activity remains limited, and it is not known whether it is required for function although studies of the yeast homolog COQ8 suggest that this is the case[52]. PG contains a reservoir of plastoquinone that is required to replenish the plastoquinone that has been damaged by reactive oxygen species formed in the photosynthetic electron transport chain. This process is under the control of ABC1K1 and is known as plastoquinone homeostasis. The *abc1k1* phenotype results from a deficiency of plastoquinone in the photosynthetic electron chain[46]. This will increase excitation pressure at PSII and cause charge recombination damaging D1[46]. Binding of urea herbicides such as DCMU (3- (3,4-dichlorophenyl)-1, 1-dimethylurea) (added in a low dose) to the PQ-binding niche Qb stabilizes[53–55] and preserves D1 from proteolytic cleavage under otherwise damaging conditions[53,54]. The binding of a low dose of DCMU to the Qb site of D1 increases the redox potential at the Qa site favoring direct recombination to the ground state and avoiding the production of highly reactive radicals $^3P_{680}$ (triplet chlorophyll) and $^1O_2$ (singlet oxygen)[56–58].

Here, we demonstrate that illumination of *var2* and *abc1k1* with continuous monochromatic red light (60 µmol m$^{-2}$ s$^{-1}$, 680 nm, "Photosystem II light" which preferentially excites PSII and creates a damaging disequilibrium between the two photosystems) impairs photosynthesis and arrests chloroplast biogenesis at the seed-to-seedling transition. This was associated with the reduced accumulation of photosynthesis-associated proteins and, surprisingly, the accumulation of incompletely processed thylakoid proteins. The red-light phenotype of *abc1k1* was partially reversed by a low dose of DCMU which rescued preprotein processing as well as chloroplast biogenesis.

## Results

### The pale green phenotype of *abc1k1* under red light is due to a defect in chloroplast biogenesis

*abc1k1* plants develop almost colorless cotyledons under red light, failing to green normally. To determine whether this is due to bleaching (defined as the loss of pigmentation), we narrowed down the time point at which cotyledon greening was affected in *abc1k1*. We germinated the plants under continuous moderate white light (WL, 80 µmol m$^{-2}$ s$^{-1}$) for increasing durations before transferring them to continuous monochromatic red light (RL, 60 µmol m$^{-2}$ s$^{-1}$, 680 nm). *abc1k1* plants, germinated for 48 h under WL and then shifted to RL for 7 days, had greener cotyledons than those germinated under constant RL. This suggested that *abc1k1* interferes with chloroplast biogenesis at the seed-to-seedling transition when exposed to RL (Fig. 1a; left-hand panel). Consistently, *abc1k1* mutant plants germinated under RL for 48 or 72 h (to compensate for the slower germination) before transfer to WL, remained compromised in cotyledon greening, and more so after 72 h (Fig. 1a; right-hand panel). Throughout the five days growth period, no greening under RL comparable to that under WL could be observed in the tiny *abc1k1* seedlings (Fig. 1b). Considering the onset of the phenotype at germination and the failure to green normally, the *abc1k1* defect under RL likely arises from the arrest of chloroplast biogenesis and not from bleaching.

To characterize the ultrastructural phenotypes of the chloroplasts contained in the Col-0 and the *abc1k1* mutants under WL and RL, transmission electron microscopy was carried out (Fig. 1c). Under WL the Col-0 wildtype contained typical chloroplasts with both stromal and stacked thylakoid membranes as well as large starch granules. Under WL *abc1k1* chloroplasts looked very similar. Under RL, Col-0 chloroplasts still showed large starch granules as well as thylakoid membranes. In comparison, colorless cotyledons of *abc1k1* contained chloroplasts that lacked the typical thylakoid membranes and starch granules but appeared vacuolated and displayed large plastoglobules.

### The *var2* (*ftsh2*) response to red light resembles that of *abc1k1*

To further investigate the *abc1k1* phenotype, a panel of Arabidopsis genotypes (Col-0, *abc1k1-1*, *abc1k1-2*, *var2*, and *sps2*) was germinated and grown for 5 days either under WL or RL. *var2* was selected because it developed a variegated phenotype similar to that of *abc1k1* under high light intensity. The solanesyl phosphate synthase mutant 2 (*sps2*) was selected because it has overall lower levels of plastoquinone and, in some aspects mimics the *abc1k1* phenotype[46].

None of the genotypes developed a visible phenotype under WL whereas RL affected all genotypes but to varying degrees (Fig. 2a). Under RL, seedlings in general were smaller and slightly paler except for the two *abc1k1* alleles and *var2*. These were almost colorless, with *var2* appearing paler than the two *abc1k1* alleles. This was assessed by chlorophyll measurements. The chlorophyll concentrations varied under WL but remained above 200 ng /mg fresh weight (Fig. 2b). Under RL, when compared to WL the chlorophyll concentrations were slightly reduced in Col-0 and *sps2* and much more severely in the two *abc1k1* alleles and *var2*. The biggest reduction was seen in *var2*. The PSII quantum yield ($\Phi_{PSII}$) under WL was lower in all mutant genotypes compared to Col-0, but under RL, it was reduced to a much higher degree in all mutants, particularly in *abc1k1* alleles and *var2* indicating a defect in photosynthesis (Fig. 2c).

As chlorophyll levels and PSII quantum yield ($\Phi_{PSII}$) were affected under RL we analyzed and compared by immunoblotting a set of photosynthesis-associated proteins representing Photosystem II (PsbA, -B, -O1, -Q, -P), Cytochrome $b_6f$ complex (PetB and -C) and Photosystem I (PsaD, -F, -N). This set was completed by plastocyanin, Light-harvesting complex b1 (Lhcb1), Light-harvesting complex a1 (Lhca1), TOC75 and phosphorylated PsbA (PsbA-P) and Lhcb1 (Lhcb1-P) (Fig. 2d). The levels of all these proteins in all genotypes were quite similar after 5 days under WL. However, after 5 days under RL their levels, apart from TOC75, were severely reduced in the two *abc1k1* alleles and *var2* when compared to Col-0 and *sps2* (Fig. 2d, Red Light). Generally, the apparent masses of the assessed

a

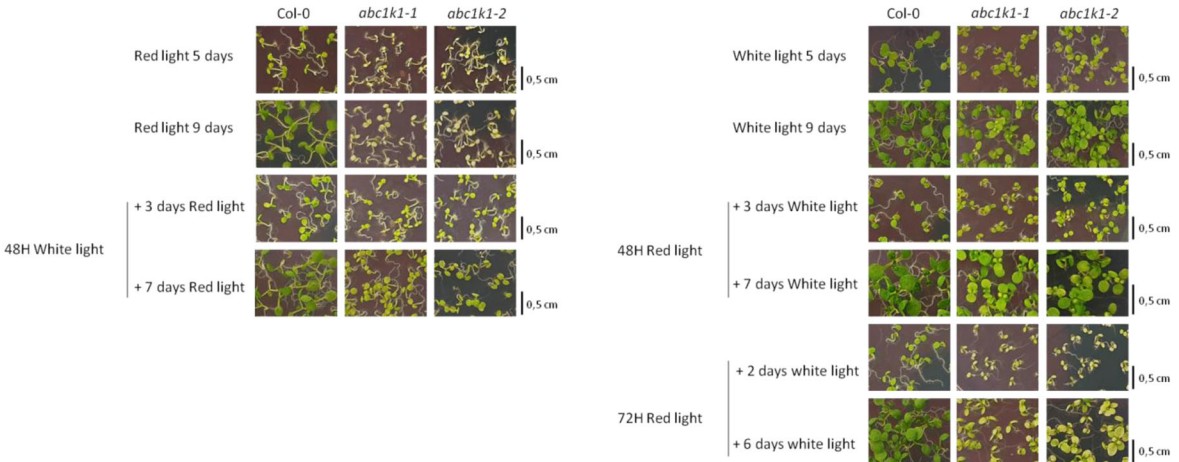

b

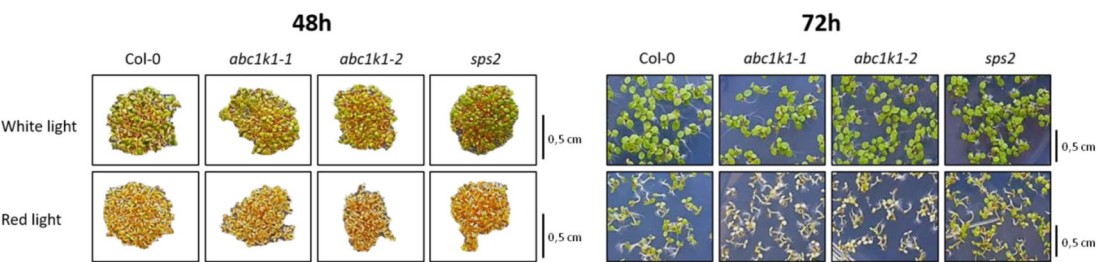

c

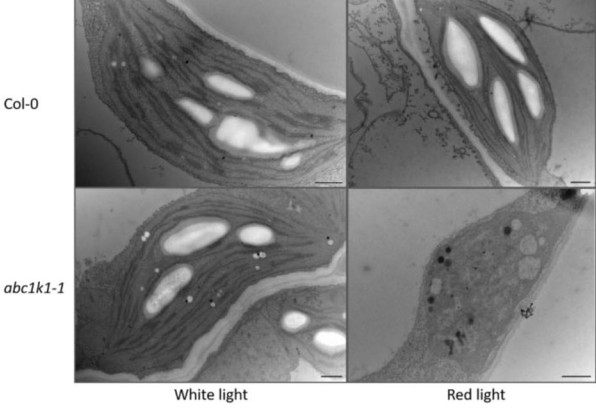

**Fig. 1 | The *abc1k1* mutant suffers from a chloroplast biogenesis defect under red light. a** Visible phenotype of 5 and 9 days old seedlings of Col-0, *abc1k1-1* and *abc1k1-2* exposed to moderate white light (left) or red light (right) until radicle emergence and then moved to the other light condition. **b** Visible phenotypes of 48 and 72 h-old seedlings of Col-0, *abc1k1-1* and *abc1k1-2* exposed to moderate white light or red light. **c** Transmission electron microscopy (TEM) images of Col-0 and *abc1k1-1* cotyledon cells after 5 days of moderate white light or red light. Scale bar: 50 nm.

a

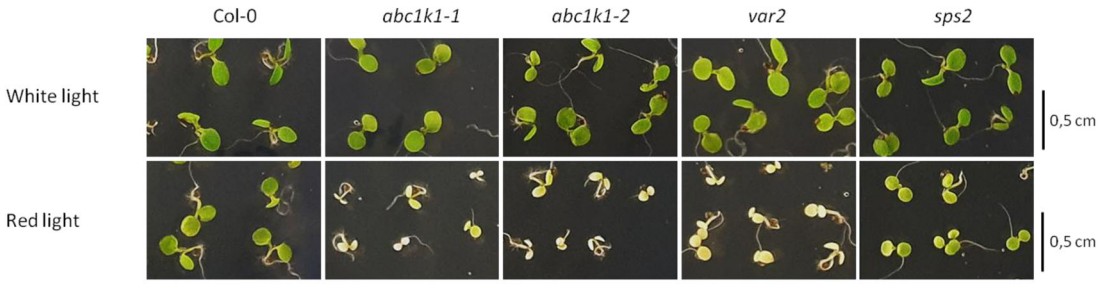

b

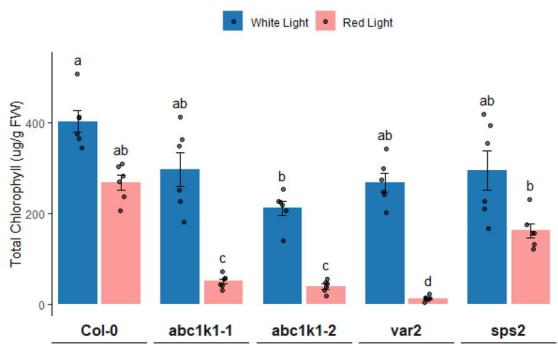

c

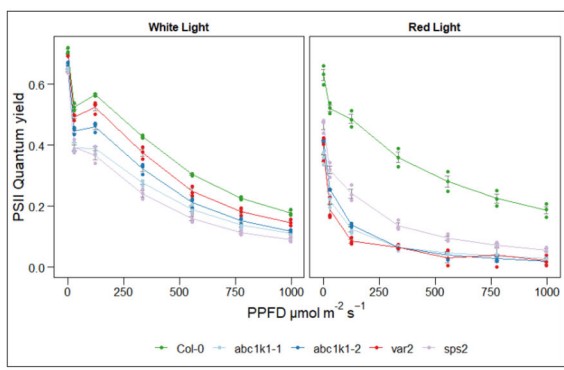

d

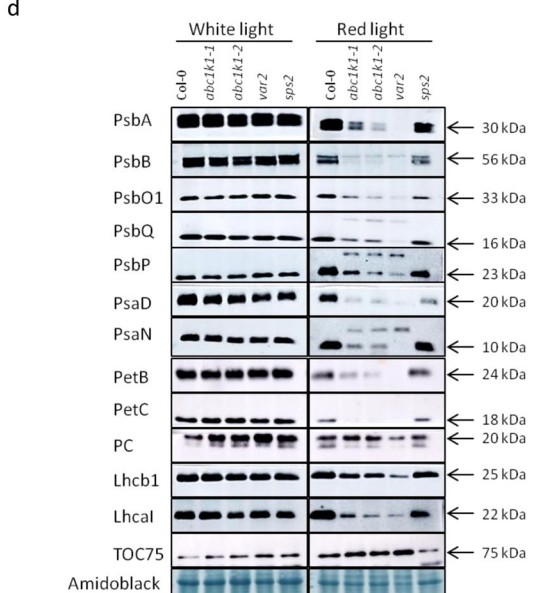

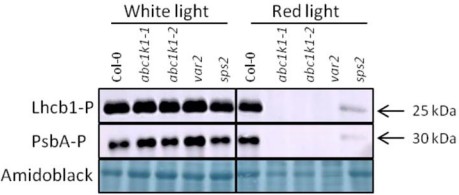

**Fig. 2 | *abc1k1* and *var2* have similar defects in chloroplast biogenesis under red light.** Col-0, *abc1k1-1*, *abc1k1-2*, *var2*, and *sps2* seedlings were grown for 5 days under moderate white light or red light on standard 0.5× MS media. **a** Representative image of the phenotypes. **b** Chlorophyll levels in seedlings 5 days after germination, the bar plot shows the total chlorophyll concentration. The letters identify statistically different groups obtained by an estimated marginal means post-hoc test (alpha 0.05). Error bars represent the standard error of the mean ± SD (*n* = 6). **c** PSII quantum yield ($\Phi_{PSII}$) of Col-0 and mutants. Error bars represent the standard error of the mean ± SD (*n* = 3). **d** Representative immunoblots of the different genotypes showing the accumulation of proteins of: photosystem II (PsbA/D1, PsbB, PsbO1, PsbQ, PsbP), photosystem I proteins (PsaD, PsaN), cytochrome $b_6f$ (PetB, PetC), as well as plastocyanin (PC), light-harvesting complex b1 (Lhcb1), light-harvesting complex a1 (LhcaI), TOC75, and phosphoproteins Lhcb1-P and Psba-P.

proteins were the same as in Col-0 under WL. However, for PsbQ, PsbP, and PsaN the immunoblots revealed an additional band of unknown nature and higher molecular mass.

### Proteome-wide analysis revealed diminished photosynthetic proteins under red light in Col-0 and *abc1k1* and *var2* mutants

The immunoblotting experiments were in agreement with the profound perturbation of photosynthesis in *abc1k1* and *var2* under RL. To gain further insight into differences at the level of the total proteome, protein extracts were prepared from 5 days old seedlings of Col-0, *abc1k1-1*, *-2*, *var2*, *sps2*. Samples were digested with trypsin and analyzed by liquid-chromatography-mass spectrometry with data-independent acquisition (DIA) on a trapped ion mobility mass spectrometer. The analysis led to the identification and quantification of 9086 protein groups (Supplementary Data 1). Principal component analysis (Fig. 3a) revealed a major first component accounting for 47.4% of the variance, which clearly discriminated light conditions and genotypes. A minor second component (8.9% of variance) was experiment-linked. In the main first component, all genotypes clustered together in WL, while they were clearly separated in RL. As expected, Col-0 RL samples were the most similar to white light samples, while *abc1k1* RL and *var2* RL displayed increasingly large differences in their proteomes relative to a WL phenotype. Statistical tests confirmed these trends. Less than 5 proteins were significantly different between genotypes in WL conditions, while several hundred proteins passed the test under the RL conditions (Table 1). All RL samples showed strong differences when compared to their WL controls, with almost one-third of the proteome impacted by RL in the *abc1k1* genotype and almost half in *var2*. In *abc1k1*, almost 1000 proteins (965) were significantly different in RL conditions in comparison to the Col-0 RL control (Table 1). Gene Ontology (GO) annotation enrichment analysis on the fold changes between conditions highlighted photosynthesis-associated terms as the most strongly impacted by RL but also by the genotype. Components of Photosystem I and II, the electron transport chain, and the oxygen-evolving complex had the strongest scores in this analysis (Supplementary Data 1). KEGG functional categories summarize well broad global changes (Table 2), showing that photosynthesis-associated proteins, while decreased in Col-0 in RL, are even more strongly depleted in *abc1k1* and *var2*. Levels of ribosomal proteins as a class were quantitatively unaffected in Col-0 in RL but decreased in both mutants. Simultaneously, the proteasome was upregulated in all RL conditions but significantly more in *abc1k1* and *var2* than in Col-0, suggesting altered proteostasis in both mutants (Table 2). Interestingly, *abc1k1* and *var2* also displayed increased levels of peroxisomal proteins, suggesting increased reliance on fatty acid oxidation as an alternative energy source in these genotypes.

A more detailed inspection of the data of the proteomes of *var2* and *abc1k1* vs. Col-0 in RL showed that components of Photosystem I and II, the electron transport chain and the oxygen-evolving complex had the strongest scores in this analysis and were strongly depleted in the mutants, significantly more than the average of all chloroplast proteins (Fig. 3b, Supplementary Data 1).

Based on the enrichment of GO annotations related to photosynthesis, the workflow was adapted, and further analyses were directed towards specific proteins containing Psa (for Photosystem I, termed PsaX proteins), Psb (for Photosystem II, termed PsbX proteins) and Pet (for cytochrome $b_6f$, termed PetX proteins) in their gene names, defining a group of 35 proteins. The volcano plot of the comparison between *abc1k1* and Col-0, both under RL, illustrates that 25 out of 35 photosynthesis-associated proteins accumulated to a significantly lesser degree in *abc1k1* (Fig. 3c). Proteome composition thus correlates strongly with the previous observations regarding the phenotypes under different light conditions (Fig. 2a, b). All PsaX, PsbX, and PetX proteins (with the exception of PsbW) trended lower under RL in all genotypes but particularly so in *abc1k1* and *var2* (Supplementary Data 1).

Interestingly, in a hierarchical clustering of protein abundances in this group, the Col-0 samples in RL appeared more similar to the WL samples,

while the *abc1k1* and *var2* RL samples clustered separately (Fig. 4a, b). This contrasts with the whole proteome data in which, in the same analysis, Col-0 RL samples are clearly separated from WL samples. This observation may be explained by the activity of red light signaling pathways in Col-0 that are overruled by retrograde plastid-to-nucleus signaling pathways triggered by the photosynthetic defects in *abc1k1* and *var2* and particularly affect GO annotations related to photosynthesis.

### Accumulation of incompletely processed forms of photosynthesis-associated thylakoid proteins in *abc1k1* and *var2* after red light

The higher molecular mass bands of several photosynthesis-associated proteins in the immunoblots suggested the presence of additional peptide sequences in the corresponding proteins. We analyzed more closely the set of peptides identified and quantified in our mass spectrometry data for the 35 PsaX, PsbX, and PetX proteins mapped by a total of 259 peptides (Supplementary Data 1 and 2).

As expected from the overall trend at the protein level, the MS signal for most peptides from PsaX, PsbX, and PetX decreased in intensity in RL compared to WL as well as in mutants compared to Col-0 controls. Indeed, in RL conditions 156 peptides out of 259 had a statistically significantly different intensity in *abc1k1* when compared to Col-0 (196 for *Var2*) (Fig. 5a). While most of these peptides had a much lower signal, 12 peptides showed a markedly opposite trend, with a higher intensity in *abc1k1* than in the control (Fig. 5b). A similar trend was observed for the *var2* mutant. Plotting the peptide intensity ratio between *abc1k1* and Col-0 under RL against the peptide position in the amino acid sequence showed that all 12 peptides mapped near the N-termini of the corresponding protein (Fig. 5c), approximately between position 20 and 100. All these peptides belonged to seven thylakoid lumen proteins: four in Photosystem II (PsbO1, PsbO2, PsbP, PsbQ, and PsbT) and two in Photosystem (PsaN and PsaF). Furthermore, all peptides from these five proteins that were located C-terminally to position 100 showed clear decreases in *abc1k1*, coherent with the general trend of the protein abundance (Fig. 6b, showing PsbQ2 as an example).

Comparison with the Plant Proteome Database (PPDB, http://ppdb.tc.cornell.edu/) and SUBA5 (subcellular localization database for *Arabidopsis* proteins, https://suba.live/) revealed that the 12 identified peptides near the N-terminus corresponded to predicted targeting peptides that were either responsible for translocation across the chloroplast envelope membranes (three peptides; PsaN (amino acids 27–39), PsbO1 (37–46)) and PsbT (31–43) or across the thylakoid membrane (9 peptides; PsaF (47–76), PsbO1 (68–90), PsbO2 (67–89), PsbP1 (57–71, 72–88, 72–90), PsbQ2 (62–78, 79–88), PsbT (53–68) using either the TAT or SEC pathways (Figs. 5a, b, 7, Table 3).

An in-depth analysis of the proteomics data revealed 14 additional proteins that behaved in similar albeit less pronounced fashion: Amongst the 71 thylakoid lumen proteins[15] the accumulation of such peptides under RL was also observed in 2 other proteins (AT5g45680; AT4g24930), which were also of lower abundance (Fig. 6a). Using the same criteria, an additional 12 proteins were identified in a list of more than 1000 chloroplast proteins with known or predicted chloroplast transit and luminal targeting peptides (according to http://ppdb.tc.cornell.edu/ and https://suba.live/). Namely, these were six chlorophyll-binding proteins (AT4G10340; AT5G01530; AT3G54890; AT2G34420; AT5G54270; AT1G15820), two subunits of the ATP-synthase (AT4G32260; AT4G04640) and three stromal proteins (AT3G01500; AT3G54050; AT5G23060). Thus, with the exception of the three stromal proteins, all others were thylakoid components (Table 3).

### The *abc1k1* phenotype can be restored with a small dose of DCMU

We wished to address whether the pale green phenotype of *abc1k1* was linked to the lack of Qb occupancy by plastoquinone in D1 (resulting in damaging charge recombination) and whether it could be reversed (including the cleavage of preproteins). To do so, wild type, *abc1k1* and *var2*

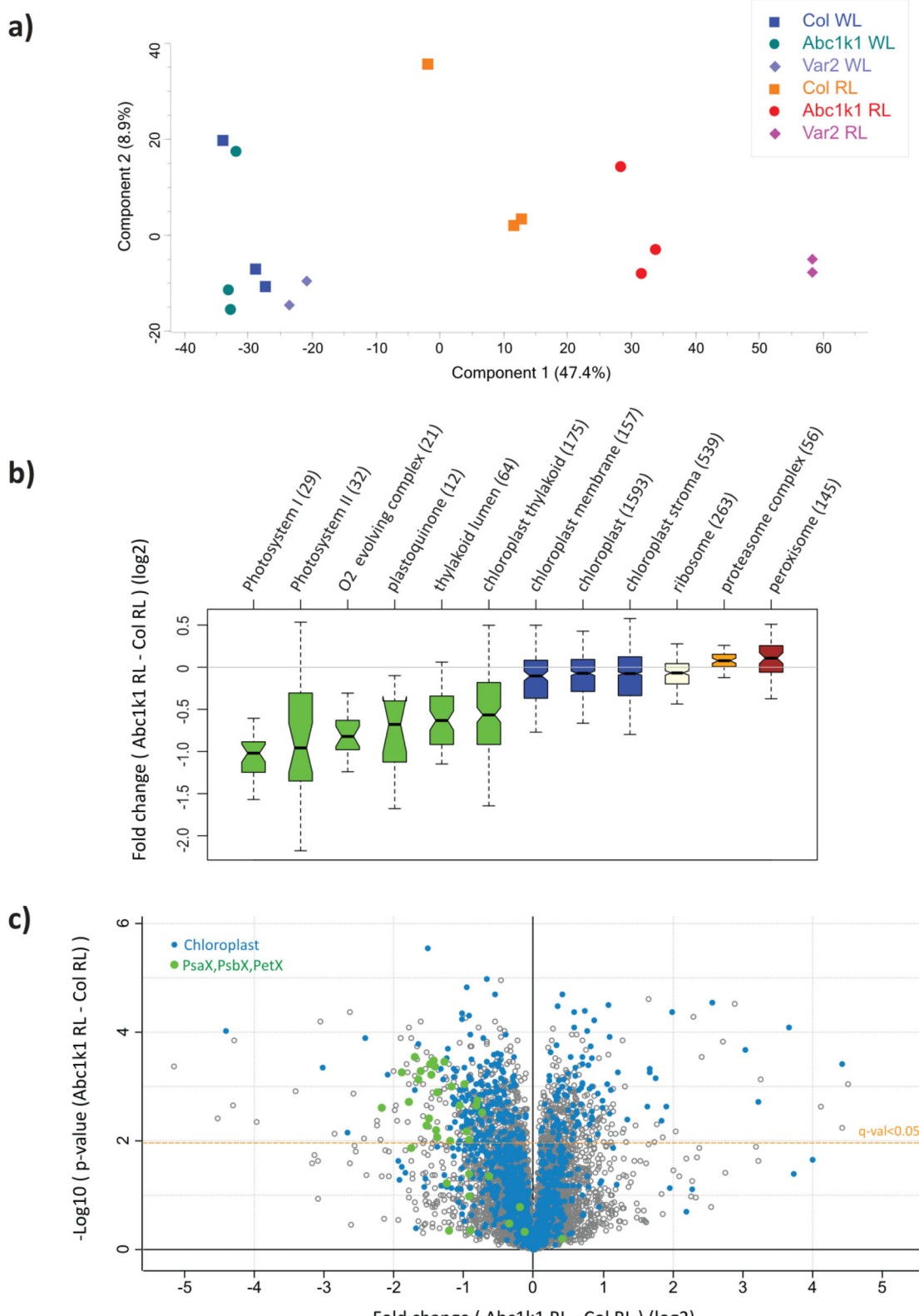

**Fig. 3 | Comparison of protein profiles in genotypes exposed to moderate white or red light. a** Plots derived from an untargeted principal component analysis (PCA) of proteome compositions of 5-day-old seedlings of Col-0, *abc1k1-1*, *var2* under moderate white light (WL) and red light (RL). **b** Comparison of median log2 fold changes of *abc1k1-1* compared to Col-0 under red light (RL) of selected gene ontology (GOCC, cellular compartments) categories. Numbers of proteins associated with each category are in brackets. **c** Volcano plot comparing protein contents of Col-0 *and abc1k1-1* under red light (RL). Proteins with −log10(*p*-val) > 1.98 passed the FDR-adjusted threshold (black). Values for 35 proteins belonging either to Photosystem I, Photosystem II or linked to the cytochrome b$_6$f complex are shown as green filled circles, chloroplast proteins are in blue.

**Table 1 | Numbers of statistically significant proteins in the mass spectrometry-based global comparisons**

| Comparison | Proteins passing *T*-test (Benjamini–Hochberg correction, adj. *p*-val < 0.05) | Percentage of total |
|---|---|---|
| *var2* WL–*abc1k1* WL | 5 | 0.06 |
| *var2* WL–Col WL | 2 | 0.02 |
| *abc1k1* WL–Col WL | 1 | 0.01 |
| *var2* RL–*var2* WL | 4363 | 48.02 |
| *abc1k1* RL–*abc1k1* WL | 2936 | 32.31 |
| Col RL–Col WL | 1200 | 13.21 |
| *var2* RL–Col RL | 974 | 10.72 |
| *abc1k1* RL–Col RL | 965 | 10.62 |
| *var2* RL–*abc1k1* RL | 222 | 2.44 |

The dataset contained 9086 proteins quantified in a minimum of three replicates under at least one condition.

**Table 2 | Medians from 1D enrichment of selected KEGG categories**

| KEGG Term | Group size (number of proteins) | Median Col RL - Col WL | Median *abc1k1* RL - Col RL | Median *var2* RL - Col RL |
|---|---|---|---|---|
| Photosynthesis | 56 | −0.73 | −1.19 | −2.13 |
| Ribosome | 178 | 0.05 | −0.13 | −0.11 |
| Proteasome | 52 | 0.16 | 0.11 | 0.18 |
| Peroxisome | 52 | NS | 0.22 | 0.26 |

The results were filtered by adjusted *p*-value (Benjamini and Hochberg method) with a threshold of 0.05. NS = not statistically significant.

plants were grown under RL in the presence of a small dose of DCMU (12.5 nM) that is able to occupy the Qb site, thereby diminishing charge recombination[56–58] and preventing D1 degradation[53,54]. The small dose of DCMU indeed improved the phenotype in *abc1k1* under RL but not that of *var2* (Fig. 8a). The chlorophyll level increased in DCMU-treated *abc1k1* plants under red light (Fig. 8b). Notably, the D1 protein was increased and the higher mass bands of PsbP, PspQ, and PsaN in *abc1k1* were decreased in DCMU treated *abc1k1* plants but not in DCMU treated *var2* plants under RL as assessed by immunoblot (Fig. 8c). The results indicate that the low dose of DCMU partially rescued *abc1k1* but not *var2*.

## Discussion

A plant's path to photoautotrophic growth during the first three days of de-etiolation involves an initial photosynthesis establishment phase followed by one of chloroplast proliferation and growth[59]. But, whether photosynthesis is required during the initial stages of germination when seed reserves are still available has not been clarified. Here, we used two mutants, *abc1k1* and *var2* that have conditional, 680 nm monochromatic red-light (RL) sensitive phenotypes. This so-called "PSII light" preferentially excites PSII, unbalances the excitation equilibrium between the two photosystems and may inflict damage on PSII. This effect is aggravated by the mutants used in this study. They have the advantage of being conditional and are affected by accessory factors but not in essential components of the photosynthetic electron transport chain[48]. *abc1k1* lacks a sufficient supply of photoactive plastoquinone in the electron transport chain, which can lead to charge recombination and damage at the reaction center of PSII whereas *var2* is unable to repair this kind of damage at the PSII reaction center. This renders the mutants highly sensitive to RL and allows them to "switch off" photosynthesis using RL. This was evident in the reduction of the PSII quantum yield ($\Phi_{PSII}$). The mutant plants lacked chlorophyll (Fig. 2b), photosynthesis-associated proteins, and normally developed chloroplasts indicative of a failure of chloroplast biogenesis under RL. Notably, wildtype plants also suffered under RL and their proteome was distinct from that under WL, in particular gene ontology annotations of photosynthesis-

related terms were depleted. Not surprisingly, this was far more pronounced in the *abc1k1* and *var2* mutants where proteins belonging to photosynthesis-related gene ontology-terms were reduced by about 50%. In comparison, those belonging to general chloroplast-related gene ontology terms were much less reduced (around 10%) pointing to a photosynthesis-specific defect rather than a global chloroplast one. The general chloroplast-related gene ontology terms encompass many processes, and based on the overall limited reduction of protein abundances these would not necessarily be expected to be dramatically altered.

Nevertheless, the question may be asked whether it would not be possible to carry out photosynthesis with just 50% of component levels and whether further defects may underlie the observed phenotypes under RL. Along these lines, immunoblotting revealed higher molecular mass bands of several photosynthesis-associated proteins under RL suggesting that their targeting information had not been properly processed and cleaved from their preproteins forms. To address this hypothesis, we mined the mass spectrometry data to look for peptides that were near the N-termini and upregulated under RL. The analysis highlighted a set of seven partially processed, essential pre-proteins belonging to the oxygen-evolving complex of PSII (PsbO1, -O2, -P, -Q, and -T) and the plastocyanin docking site of PSI (PsaF, -N). In several cases, the identified peptides overlapped the known N-termini of the processed proteins in wild type (Fig. 7). This clearly demonstrated that the peptides belonged to preproteins and not to "free" targeting peptides that had already been cleaved from their preprotein.

In-depth analysis revealed 14 additional proteins (9 thylakoid, 2 thylakoid lumen, and 3 stromal) proteins showing similar patterns with regard to targeting peptide accumulation but to a lesser extent. Notably, the six chlorophyll-binding proteins and two ATP-synthase subunits are also thylakoid proteins albeit not lumenal ones. Overall, the proteomics data point to a specific defect at the level of the thylakoids rather than the chloroplast as a whole.

The identified peptides near the N-termini were upregulated under RL and behaved contrary to the general trend of strong downregulation (Fig. 5a, b). While it is difficult to directly compare peptide intensities, the peptides derived from targeting sequences accumulated to significant and high proportions compared to the peptides derived from mature portions of specific photosynthesis-associated components. Western blotting confirmed these observations as the preprotein bands appeared to be of similar strength as the mature ones.

The subset of seven partially processed pre-proteins of the oxygen-evolving complex and the plastocyanin docking site are localized in the thylakoid lumen and are part of the thylakoid lumen proteome of 71 proteins. Amongst the 71 thylakoid lumen proteins, only two others, less abundant, showed comparable but less pronounced behavior, whereas no other thylakoid lumen proteins produced peptides derived from targeting sequences. This indicated that the large majority of thylakoid lumen proteins had been normally processed and that the phenomenon was around 90% specific. It also indicates that the thylakoid processing peptidase PLSP1

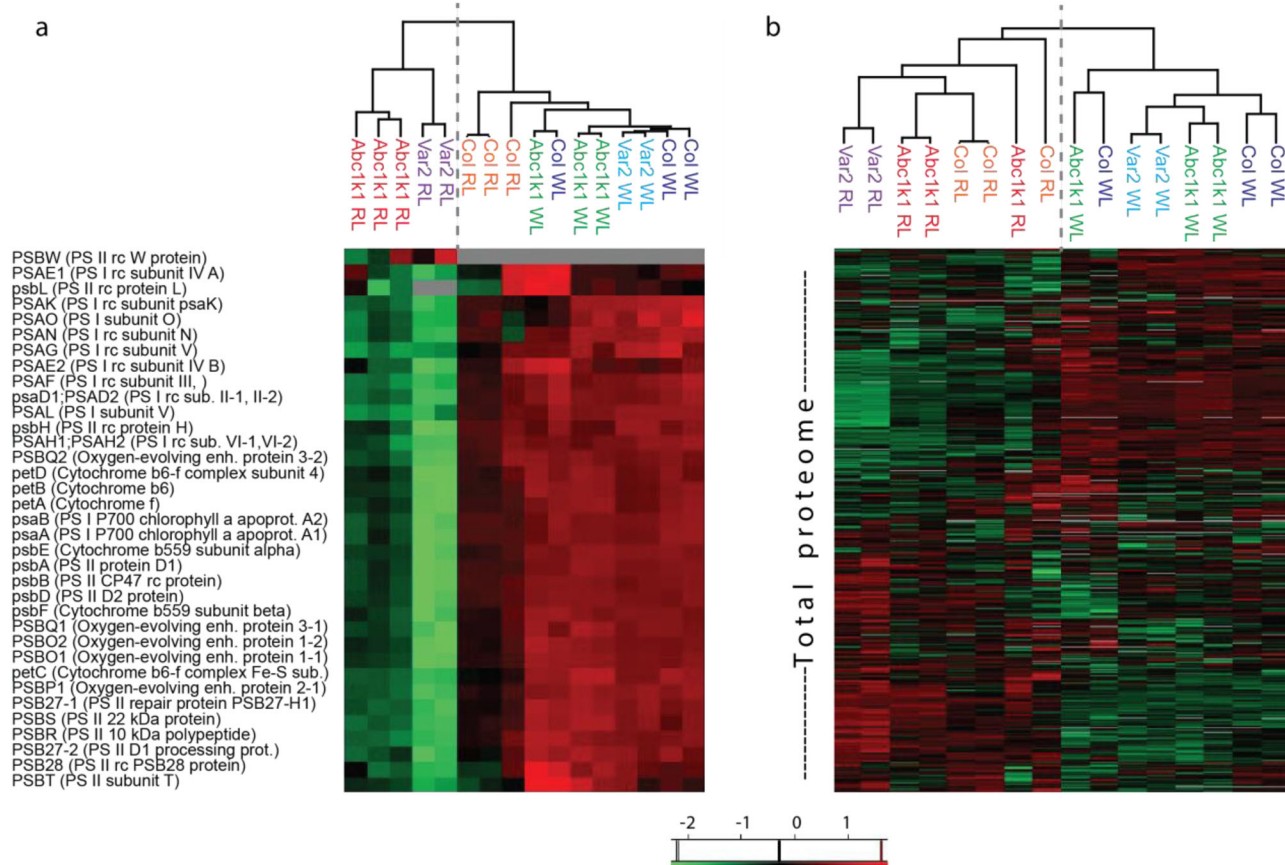

**Fig. 4 | Hierarchical clustering of abundances of 35 photosynthesis-associated proteins and total proteomes in genotypes under red compared to moderate white light.** MS-derived protein quantities were z-scored before clustering. **a** 35 proteins of the photosynthetic machinery. **b** Total proteomes (7281 protein groups). WL = moderate white light, RL = red light. Separation between the main branches of the trees is indicated by a dashed line.

was active. This was confirmed for the TOC75 and plastocyanin reference PLSP1 substrates that appeared to be normally processed. PLSP1 is essential, and its knockout results in an albino phenotype underscoring the indispensable requirement for preprotein processing for functional photosystem assembly[30,60]. In in vitro thylakoid translocation experiments, higher molecular mass bands were also observed and corresponded to partially translocated preproteins indicating that transit peptide processing only occurs once translocation is complete[30]. Although higher molecular mass forms were observed in the *plsp1* background, proteome-wide profiling of targeting peptides was not carried out, so the extent of the processing defect remains unknown[30,31,60]. However, from the experiments in the present study, limited by the tiny size of the seedlings, it cannot be concluded whether the partially processed preproteins resulted from non-translocation or non-processing by PLSP1. This represents a gap in knowledge that we plan to address in the future.

Surprisingly maybe, *abc1k1* and *var2* yielded the same set of partially processed proteins and peptides accumulating under RL, and those were also observed under RL in the wild type but to a lower degree. This suggested that they are inefficient transport substrates. The seven proteins of the oxygen-evolving complex and the plastocyanin docking site are amongst the most abundant proteins of the chloroplast according to our proteome analysis (Supplementary Data 1), which may compound the defect compared to less abundant thylakoid lumen components, even though two such proteins were observed.

Substrates of both the SEC (PsaF, PsbO1, PsbO2), TAT (PsaN, PsbP1, PsbQ2, PsbT,) and SRP (chlorophyll-binding proteins) pathways were represented. The ATP and GTP for the SEC and SRP pathways and the proton gradient required to drive that of TAT are products of photosynthetic activity. We therefore hypothesize that even at an early

developmental stage, when seed reserves are still available, photosynthesis may be necessary to support the chloroplast SEC, SRP, and TAT pathways. It is interesting to note that in the mutants under red light GO-terms related to peroxisomes, which contribute to energy production from seed reserves, were upregulated. This may allow the power of the TOC-TIC import pathway across the envelope membranes given that general, non-photosynthesis-associated proteins accumulated and were processed nearly as in wild type. The RL treatment of *abc1k1* and *var2* affects the integrity of the D1 protein and disrupts photosynthesis at the onset of germination. This was partially reversed by a low dose of DCMU in *abc1k1* but not in *var2* which could be explained by the normal availability of plastoquinone and occupancy of the Qb site of D1 in *var2* but not in *abc1k1*. Low-dose DCMU also reversed the partial preprotein processing defect in *abc1k1* (but not *var2*) confirming the association of processing with photosynthetic activity. Together, the findings in this report support the notion that the photosynthetic defects in both mutants originated at the D1 protein and resulted in failure to sufficiently energize the nascent thylakoid membrane. Concomitantly, correct processing of several thylakoid proteins essential for photosynthetic electron transport was disrupted. We hypothesize that the onset of photosynthesis and processing of the observed set of thylakoid preproteins are mutually dependent constituting a critical step in chloroplast biogenesis.

## Methods
### Plant materials, growth conditions, and treatments
Arabidopsis thaliana wild-type is var. Columbia-0 (Col-0). Two ABC1K1 T-DNA insertion lines (SALK_068628, 366 SALK_130499C) were obtained from the Nottingham Arabidopsis Stock Centre (NASC, http://arabidopsis.info). The *sps2* mutant was from Gilles Basset. The *var2* (*ftsh2*) mutant was

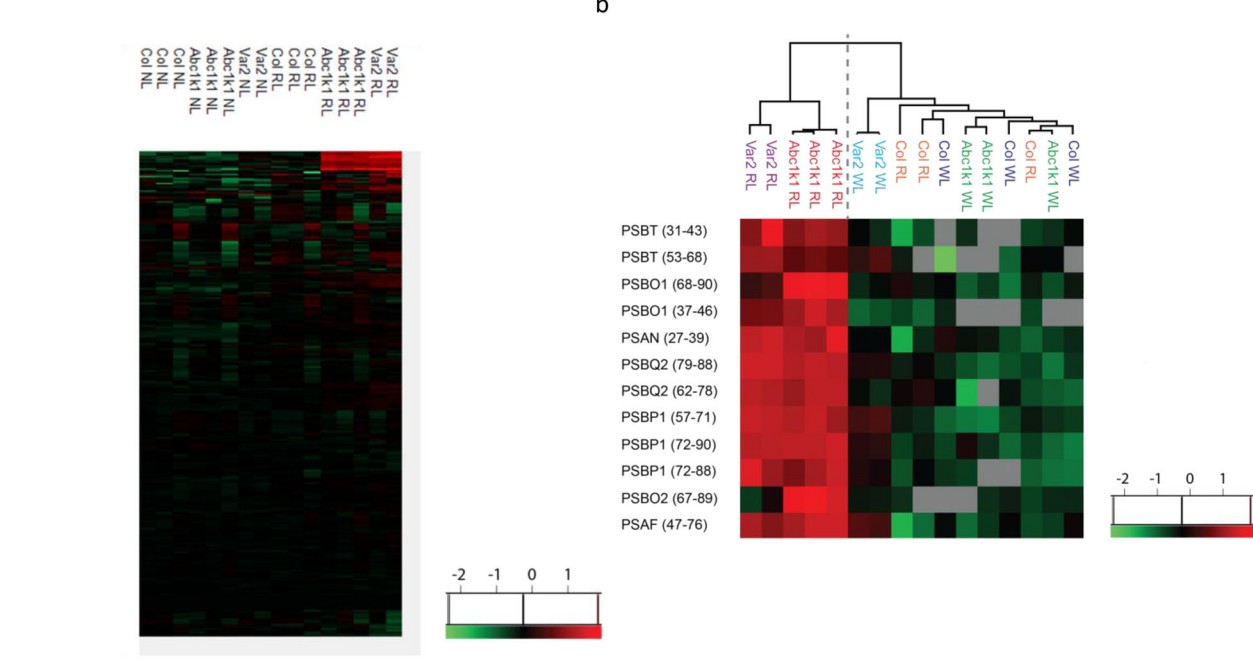

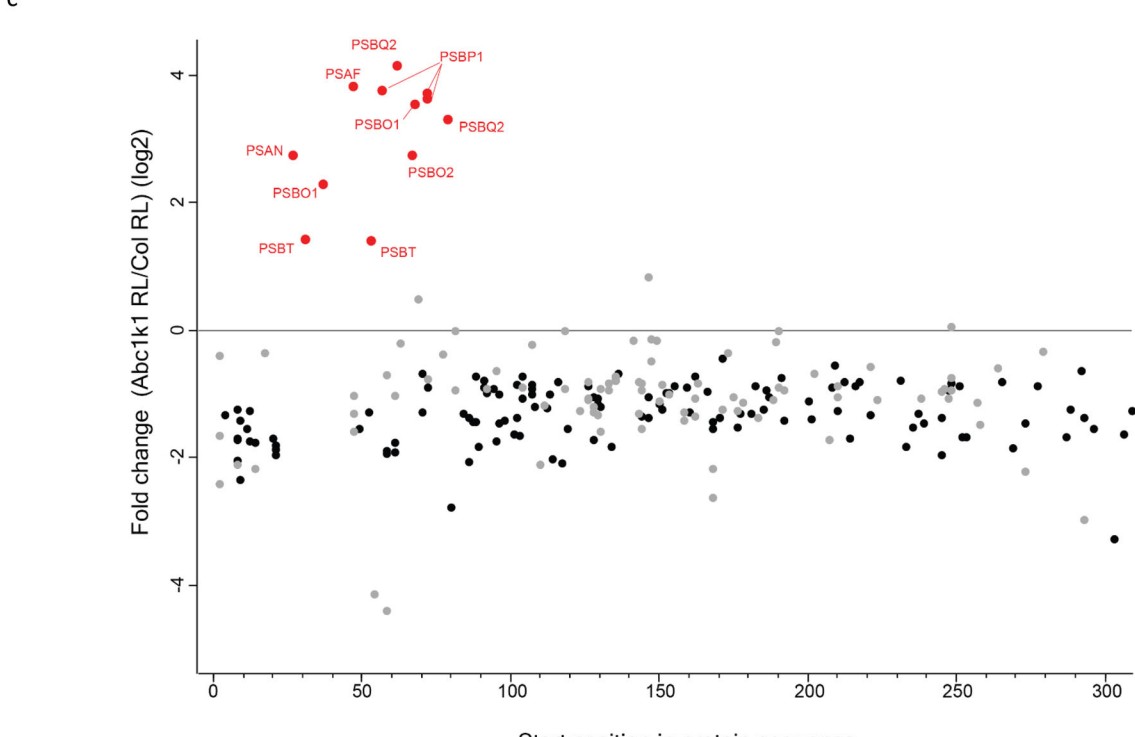

**Fig. 5 | Accumulation of specific peptides of Psa and Psb proteins under red light in *abc1k1* and *var2*. a** Heat Map showing the MS-determined intensities of all 267 peptides derived from 35 proteins of the photosynthetic machinery. **b** Heat Map showing the MS-determined intensities of 12 identified Psa and Psb peptides in Col-0, *abc1k1-1* and *var2* under moderate white light (WL) or red light (RL). MS signal intensities were *z*-scored before plotting. **c** Plot showing the accumulation of peptide fragments in *abc1k1-1* under red light (RL) compared to Col-0 control, as a function of position in the sequence. Peptides with statistically significant signal intensities are shown with black (more abundant in Col-0), respectively, red (more abundant in *abc1k1-1*) circles. Points for positions beyond AA 300 were consistently below zero. The 12 peptides labeled in red are the same as shown in panel (**a**).

from Wataru Sakamoto. Sterilized seeds were spread on 0.5× MS plates with or without 12.5 nM of DCMU (3-(3,4-dichlorophenyl)-1, 1-dimethylurea), and were placed in the dark for 24 h at 4 °C. Plates were moved to 22–24 °C and exposed to moderate white light (80 μmol m$^{-2}$ s$^{-1}$) for 1 h and kept under continuous moderate white light (80 μmol m$^{-2}$ s$^{-1}$) for 5 days or moved to continuous monochromatic red light (60 μmol m$^{-2}$ s$^{-1}$, 680 nm). 5-days-old seedlings were collected under the respective light, immediately frozen in liquid nitrogen and stored at −20 °C.

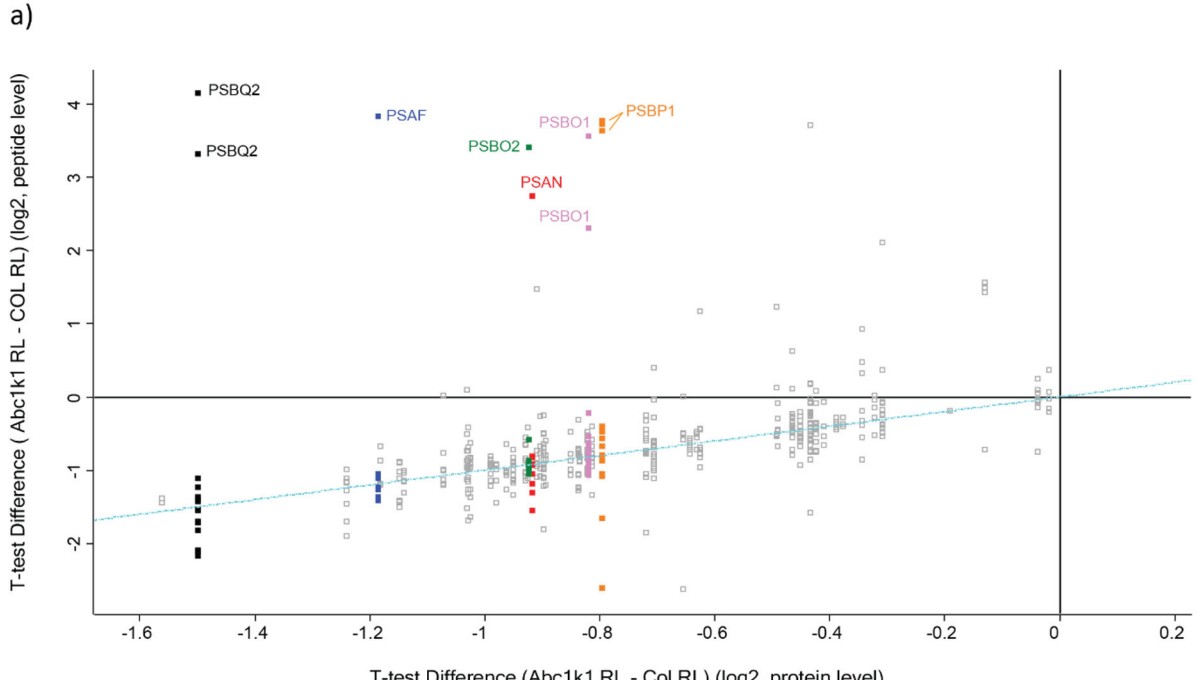

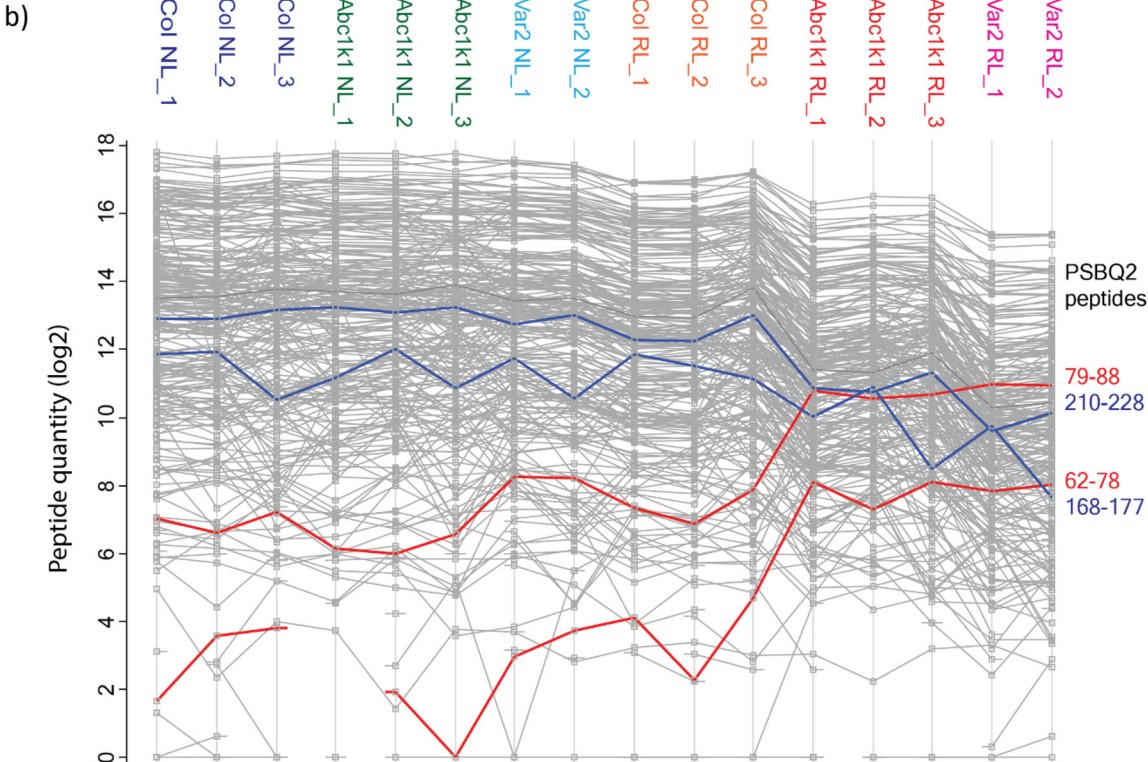

**Fig. 6 | Peptide intensities across genotypes under red and moderate white light. a** Plot showing the accumulation of peptide fragments in *abc1k1-1* compared to Col-0 control as a function of the difference of the protein levels in *abc1k1-1* compared to Col-0 all under red light (RL). The plot shows the data for 71 thylakoid lumen proteins as defined by Farci et al. (https://doi.org/10.3389/fphgy.2023.1310167). Selected sets of peptides for the same protein are color-coded and appear aligned vertically, having the same overall protein ratio. **b** Red trace peptides correspond to the thylakoid targeting sequence of PSBQ2 (AVFAEAIPIK (79–88); SVIGLVAA-GLAGGSFVK (62–78)). In blue two peptides representing the mature portion of the same protein (YDLNTVISAK (168–177); SSPDAEKYYSETVSSLNNVLAK (210–228)). Shown are MS-determined peptide quantities.

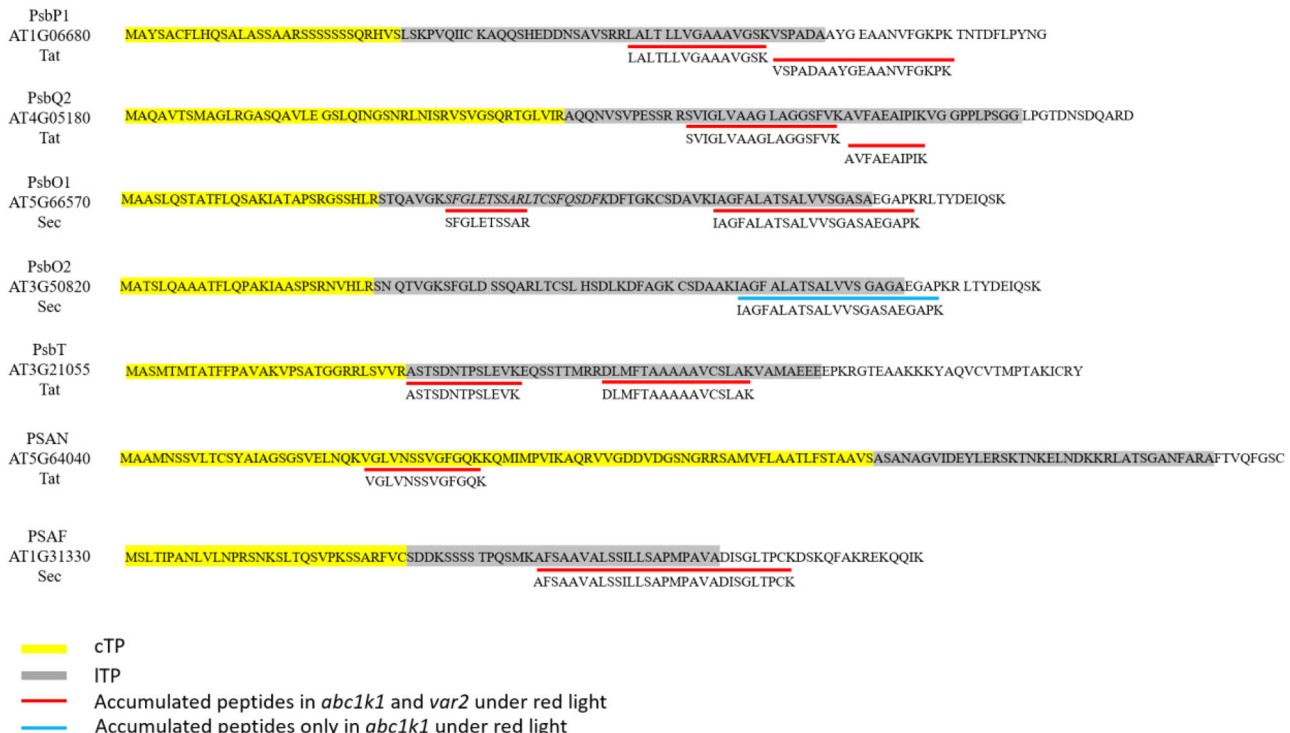

**Fig. 7 | N-terminal amino acids sequence of PsbP1, PsbQ2, PsbO1, PsbO2, PsbT, PsaN, and PsaF.** cTP (chloroplast transit peptides) are highlighted in yellow, lTP (lumen transit peptides) are highlighted in gray, positions of peptides accumulated under red light in *abc1k1-1* and *var2* are underlined in red and a peptide accumulated only in *abc1k1* under red light is underlined in blue. Positions of cTP and lTP are based on Plant Proteome Data Dase (PPDB) predictions. The position of cTP and lTP for PsbT was based on a comparison with the cotton homolog[71].

## Immunoblot analyses

Frozen 5 days old seedlings were ground in lysis buffer (100 mM Tris–HCl pH 8.5, 2% SDS, 10 mM NaF, 0.05% of protease inhibitor cocktail for plants (Sigma)) using a micro-pestle in a 1.7 ml microtube. Extracts were incubated at 37 °C for 30 min and centrifuged for 15 min at 16,000×*g* at room temperature. Protein was determined using the Pierce BCA protein assay kit (Thermo Scientific, cat. no. 23225). Proteins were precipitated using chloroform–methanol and dissolved in sample buffer (50 mM Tris–HCl pH 6.8, 100 mM Dithiothreitol, 2% SDS, 0.1% Bromophenol Blue, 10% Glycerol) at a final concentration of 1 μg/μl. After denaturation for 10 min at 65 °C, aliquots of 5 μl were separated by SDS–PAGE on 16% polyacrylamide gels. Proteins were blotted to a nitrocellulose membrane for immunoanalysis using the following antibodies: anti-D1 (PsbA) (Agrisera, AS05 084), 1/5000, anti-PsbB (Agrisera, AS04 038), 1/5000, anti-PsbO1 (Agrisera, AS14 2824), 1/5000, anti-PsbQ (Agrisera, AS06 142-16), 1/1000, anti-PsbP (Agrisera, AS06 142-23), 1/3000, anti-PsaD (Agrisera, AS09 461), 1/5000, anti-PsaN (Agrisera, AS06 109), 1/1000, anti-PetB (Agrisera, AS18 4169), 1/5000, anti-PetC (Agrisera, AS08 330), 1/5000, anti-plastocyanin (PC) (Agrisera, AS06 141), 1/2000, anti-Lhcb1 (Agrisera, AS09 522), 1/10,000, anti-Lhca1 (Agrisera, AS01 005), 1/5000, anti-TOC75[61], 1/1000, anti-Lhcb1-P (Agrisera, AS13 2704), 1/10,000,anti-PsbA-P (Agrisera, AS13 2669), 1/10,000. Primary antibodies were detected with horseradish peroxidase-conjμgated anti-rabbit (Merck, AP132P) antibodies. Chemiluminescent detection was done with a luminol solution (Luminol 1.25 mM, coumaric acid 0.20 mM, mixed with 0.01% $H_2O_2$ just before the reaction). Signals were measured using a CCD camera in the Amersham Imager 600 system (AmershamBiosciences, Inc.).

## Chlorophyll quantification

Total chlorophyll was extracted from 5-day-old seedlings (minimum of 20 mg of fresh weight; FW) with 10 μl per mg FW of DMF (Dimethylformamide). Extracts were centrifuged for 1 min at 16,000×*g* and stored overnight at 4 °C in the dark. Extracts were further centrifuged 3 min at 16,000×*g* and the absorbance at 664 and 647 nm was measured with a Nanodrop spectrophotometer (NanoDrop ND-1000, Witec AG). Total chlorophyll concentrations were calculated according to Porra et al.[62].

## Analyses of photosynthetic parameters

Chlorophyll fluorescence was measured with a Fluorcam MF800 (Photon System Instrument, Czech Republic, http://www.psi.cz). The actinic light for the light curve induction was from blue LEDs (470 nm). Initially, the maximum yield of PSII was measured: $\Phi max = (FM–FO)/FM$ where FM is the maximal fluorescence in dark acclimated plants, measured during a saturating light pulse, and FO is the fluorescence recorded in the dark. At each light intensity step the quantum yield of PSII $\Phi PSII = (FM'–FS)/FM'$ was estimated. FM' was the maximal fluorescence at the end of each light intensity step, FS the steady-state fluorescence at the end of each light step, and FO' the fluorescence in the dark measured after 2 s of exposition to far-red light at the end of each light phase. Three independent biological replicates composed of 20–30, 5-day-old seedlings per genotype were measured.

## Electron microscopy

Cotyledons from 5-day-old Col-0 and *abc1k1* seedlings were immersed in fixing solution [5% (W/V) glutaraldehyde and 4% (W/V) formaldehyde in 100 mM phosphate buffer pH 6.8)] overnight at 4 °C, rinsed several times in phosphate buffer, and post-fixed for 2 h with 1% (W/V) osmium tetroxide at room temperature. After two washing steps in phosphate buffer and distilled water, the samples were dehydrated in ethanol and acetone and then embedded in Spurr's low-viscosity resin (Polyscience). Ultrathin sections of 50–70 nm were cut with a diamond knife (Ultracut-E microtome-Reichert-Jung) and mounted on uncoated copper grids. The sections were post-stained with uranyl acetate and Reynold's lead citrate. Sections were analyzed with a Philips CM 100 transmission electron microscope (Philips Electron Optics BV, Eindhoven, the Netherlands).

**Table 3 | Peptides from chloroplast proteins spanning targeting sequences, that show quantitative increase relative to the total protein ratio for the (*abc1k1* RL vs. Col RL) comparison**

| Protein ratio (log2) | Peptide ratio (log2) | Gene | Peptide position | Protein description | UniProt ID | Thy Lum[a] | Peptide sequence | Araport |
|---|---|---|---|---|---|---|---|---|
| −1.50 | 4.14 | PSBQ2 | 62 | Oxygen-evolving enhancer protein 3-2, chloroplastic | Q41932 | x | SVIGLVAAGLAGGSFVK | AT4G05180 |
| −1.19 | 3.83 | PSAF | 47 | Photosystem I reaction center subunit III, chloroplastic | Q9SHE8 | x | AFSAAVALSSILLSAPMPAVADISGLTPCK | AT1G31330 |
| −1.50 | 3.31 | PSBQ2 | 79 | Oxygen-evolving enhancer protein 3-2, chloroplastic | Q41932 | x | AVFAEAIPIK | AT4G05180 |
| −0.80 | 3.76 | PSBP1 | 57 | Oxygen-evolving enhancer protein 2-1, chloroplastic | Q42029 | x | LALTLLVGAAAVGSK | AT1G06680 |
| −0.80 | 3.73 | PSBP1 | 72 | Oxygen-evolving enhancer protein 2-1, chloroplastic | Q42029 | x | VSPADAAYGEAANVFGK | AT1G06680 |
| −0.80 | 3.64 | PSBP1 | 72 | Oxygen-evolving enhancer protein 2-1, chloroplastic | Q42029 | x | VSPADAAYGEAANVFGKPK | AT1G06680 |
| −0.82 | 3.55 | PSBO1 | 68 | Oxygen-evolving enhancer protein 1-1, chloroplastic | P23321 | x | IAGFALATSALVVSGASAEGAPK | AT5G66570 |
| −0.92 | 3.41 | PSBO2 | 67 | Oxygen-evolving enhancer protein 1-2, chloroplastic | Q9S841 | x | IAGFALATSALVVSGAGAEGAPK | AT3G50820 |
| −0.92 | 2.74 | PSAN | 27 | Photosystem I reaction center subunit N, chloroplastic | P49107 | x | VGLVNSSVGFGQK | AT5G64040 |
| −0.58 | 2.86 | PDE334 | 50 | ATP synthase beta chain (Subunit II) | Q42139 | | ALSLSSATAK | AT4G32260 |
| −0.58 | 2.59 | CA1 | 16 | Carbonic anhydrase | A0A1I9LQB3 | | LLIEKEELK | AT3G01500 |
| −0.82 | 2.30 | PSBO1 | 37 | Oxygen-evolving enhancer protein 1-1, chloroplastic | P23321 | x | SFGLETSSAR | AT5G66570 |
| −0.92 | 1.69 | LHCB5 | 22 | Chlorophyll a-b binding protein CP26, chloroplastic | Q9XF89 | | SSAPLASSPSTFK | AT4G10340 |
| −0.95 | 1.51 | LHCB4.1 | 30 | Chlorophyll a-b binding protein CP29.1, chloroplastic | Q07473 | | FTAVFGFGK | AT5G01530 |
| −0.91 | 1.47 | FKBP13 | 42 | Peptidyl-prolyl cis-trans isomerase FKBP13, chloroplastic | Q9SCY2 | x | VSSDPELSFAQLSSCGR | AT5G45680 |
| −0.89 | 1.37 | LHCA1 | 26 | Chlorophyll a-b binding protein 6, chloroplastic | Q01667 | | FVSAGVPLPNAGNVGR | AT3G54890 |
| −0.60 | 1.40 | CFBP1 | 2 | Fructose-1,6-bisphosphatase 1, chloroplastic | P25851 | | AATAATTTSSHLLLSSSR | AT3G54050 |
| −0.91 | 0.94 | ATPC1 | 43 | ATP synthase gamma chain 1, chloroplastic | Q01908 | | ASSVSPLQASLR | AT4G04640 |
| −1.55 | 0.31 | LHB1B2 | 17 | Chlorophyll a-b binding protein, chloroplastic | Q39141 | | AVKPAASDVLGSGR | AT2G34420 |
| −1.03 | 0.67 | LHCB1.2 | 2 | Chlorophyll a-b binding protein 3, chloroplastic | Q8VZ87 | | AASTMALSSPAFAGK | AT1G29920 |
| −1.30 | 0.34 | CAS | 31 | Calcium sensing receptor, chloroplastic | Q9FN48 | | QVSVSLPTSTSISLLSLFASPPHEAK | AT5G23060 |
| −1.15 | 0.19 | LHCB3 | 40 | Chlorophyll a-b binding protein 3, chloroplastic | Q9S7M0 | | YTMGNDLWYGPDR | AT5G54270 |
| −1.06 | 0.21 | LHCB6 | 43 | Chlorophyll a-b binding protein, chloroplastic | Q9LMQ2 | | TLIVAAAAQPK | AT1G15820 |
| −1.03 | 0.10 | F13M23.70 | 20 | Thylakoid lumenal 17.9 kDa protein, chloroplastic | Q9SW33 | x | LLCSLQSPK | AT4G24930 |

Peptides from all chloroplast proteins (GOCC) having a protein ratio lower than −0.4 (log2), and a peptide ratio at least 1.0 (log2) higher than the protein ratio are listed.
[a]Lumen of the thylakoid, according to Farci and Schröder[15].

**Fig. 8 | DCMU treatment partially restores the phenotype of *abc1k1* under red light.** Col-0, *abc1k1*, *abc1k1-1*, and *var2* seedlings were grown 5 days under moderate white light or red light on standard 0.5× MS media supplemented or not with 12,5 nM of DCMU. **a** Representative image of the phenotype. **b** Chlorophyll levels in seedlings 5 days after germination, the bar plot shows the total chlorophyll concentration. The letters identify statistically different groups obtained by an estimated marginal means post-hoc test (alpha 0.05). Error bars represent the standard error of the mean ± SD (*n* = 3). **c** Representative images of immunoblots of the different genotypes showing the accumulation of photosynthetic proteins PsbP, PsbQ, PsaN, and PsbA.

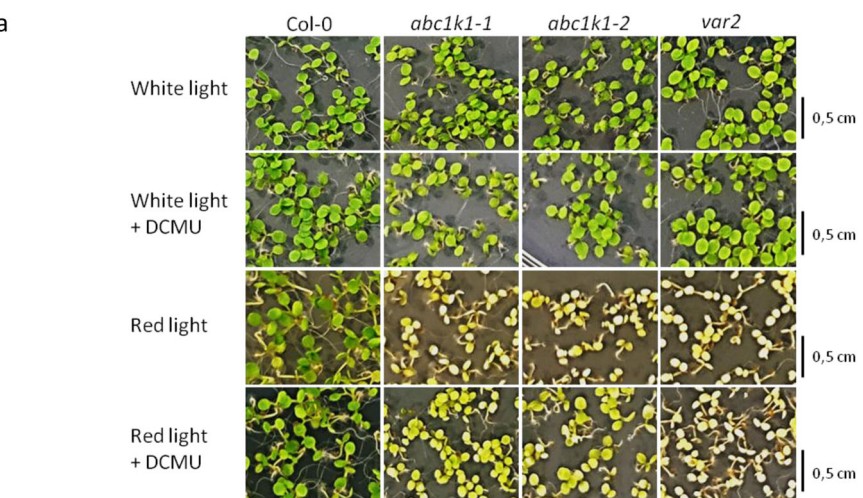

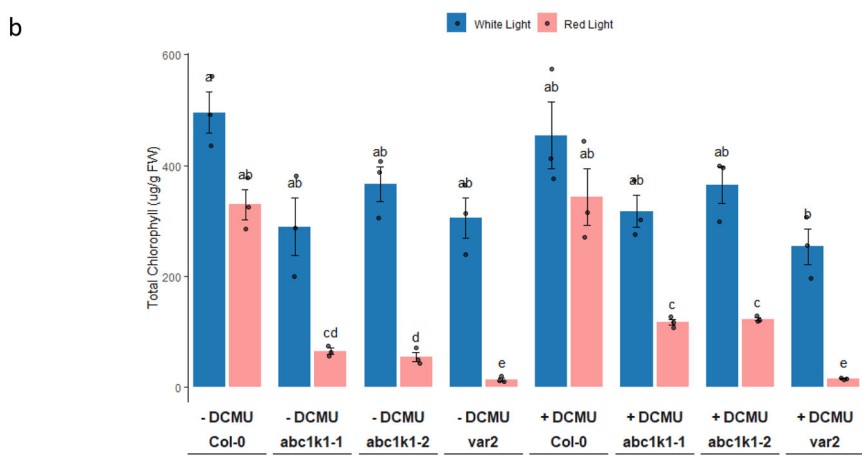

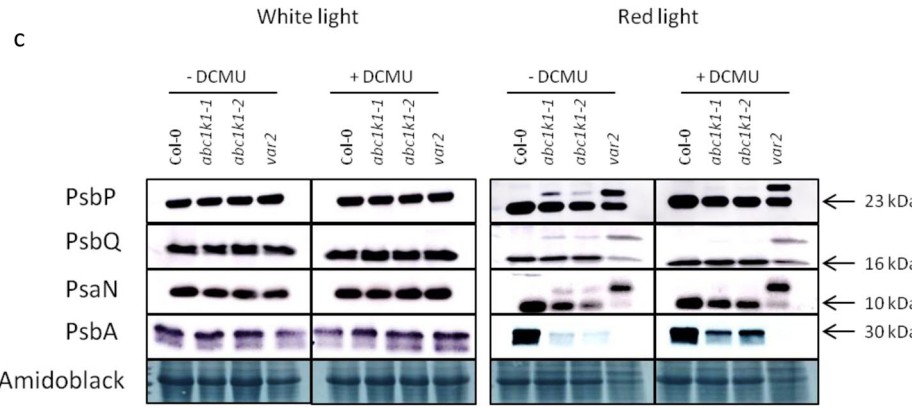

## Statistics and reproducibility

Each statistical analysis involving plant material has been done using at least three independent biological replicates, and each replicate was composed of 20–40 seedlings.

The data were analyzed with RStudio 2023.09.1 Build 494. The data were analyzed for their distribution and then processed with a Kruskal–Wallis test (Agricolae). When significant differences were found, a post-hoc analysis was performed based on a linear model and verifying its fitting with the data points, the model was then used for pairwise comparison of the estimated marginal means (emmeans), the *p*-values used to identify statistically separated groups were adjusted based on the Bonferroni correction, an alpha threshold of 0.05 was used to discriminate between groups. The groups with an alpha < 0.05 are identified with different letters in the graphs.

## Proteome analyses

Proteins were extracted as described in the subsection "Immunoblot analyses" of the "Methods" section. After the precipitation of the proteins, pellets were dried and sent to the Protein Analysis Facility in Lausanne.

## Protein digestion (SP3 protocol)

Trypsin digestion was performed according to the SP3 method[63], with minor modifications. Dried protein pellets were solubilized in 2% (w/v) SDS, 10 mM DTT, 50 mM Tris, pH 7.5 by vortexing and heating at 75 °C for 10 min. Protein concentration was determined by tryptophan fluorescence[64] and adjusted to 2 µg/µl. Reduced cysteine side chains were alkylated by reaction with iodoacetamide (final 30 mM) for 45 min at RT. An aliquot of 60 µg (30 µl) of protein was mixed with 0.6 mg of Sera-Mag beads (Cytiva product no. 451521050250), after which 1.5 volume of ethanol was added to reach a final of 60% (v/v). Tubes were agitated at 700 rpm for 15 min at RT and then placed on magnetic support for 5 min to collect the beads. The supernatant was removed, and the beads were washed five times on the magnetic support, each time with at least 400 µl of 80% ethanol, which was thoroughly removed after each step. Trypsin digestion was performed by incubating the beads in 50 µl of 100 mM ammonium bicarbonate pH 7.8 with sequencing grade Trypsin (Promega V5111) at 1/50 over protein material. After 2 h incubation at 37 °C with gentle shaking, beads were aggregated on the magnetic support. The supernatant was collected, centrifuged to eliminate bead residues, and dried by evaporation. Peptides were redissolved in 0.05% TFA and 2% MeCN at a peptide concentration of 0.2 µg/µl for LC–MS analysis (solvent A).

## Peptide fractionation for library construction

Aliquots of 5 µg of all samples were mixed to create a pool, which was manually separated into seven fractions by off-line basic reversed-phase (bRP) using the Pierce High pH Reversed-Phase Peptide Fractionation Kit (Thermo Fisher Scientific). The fractions collected were: flow-through 7.5%, 10%, 12.5%, 15%, 17.5%, and 50% MeCN in 0.1% triethylamine (~pH 10). Dried bRP fractions were redissolved in 20 µl solvent A and 2 µl were injected for LC–MS/MS analysis.

## Liquid chromatography–mass spectrometry

LC-MS/MS analysis was carried out on a TIMS-TOF Pro (Bruker, Bremen, Germany) mass spectrometer interfaced through a nanospray ion source ("captive spray") to an ultimate 3000 RSLCnano HPLC system (Dionex). Peptides were separated on a reversed-phase custom-packed 40 cm C18 column (75 µm ID, 100 Å, Reprosil Pur 1.9 µm particles, Dr. Maisch, Germany) at a flow rate of 0.250 µl/min with a 6–27% acetonitrile gradient in 92 min followed by a ramp to 45% in 15 min and to 95% in 5 min (all solvents contained 0.1% formic acid). Identical LC gradients were used for data-dependent- (DDA) and data-independent acquisition (DIA) measurements.

For the creation of the spectral library, data-dependent acquisition (DDA) was carried out on the 7 bRP fractions using a standard TIMS PASEF method[65] with ion accumulation for 100 ms for each the survey MS1 scan and the TIMS-coupled MS2 scans. The duty cycle was kept at 100%. Up to 10 precursors were targeted per TIMS scan. Precursor isolation was done with 2 Th or 3 Th windows below or above $m/z$ 800, respectively. The minimum threshold intensity for precursor selection was 2500. If the inclusion list allowed it, precursors were targeted more than one time to reach a minimum target total intensity of 20,000. Collision energy was ramped linearly based uniquely on the $1/k_0$ values from 20 (at $1/k_0 = 0.6$) to 59 eV (at $1/k_0 = 1.6$). Total duration of a scan cycle, including one survey and 10 MS2 TIMS scans, was 1.16 s. Precursors could be targeted again in subsequent cycles if their signal increased by a factor of 4.0 or more. After selection in one cycle, precursors were excluded from further selection for 60 s. Mass resolution in all MS measurements was approximately 35,000.

The diaPASEF method used the same instrument parameters as the DDA methods and was as reported previously[66]. Per cycle, the mass range 400–1200 $m/z$ was covered by a total of 32 windows, each 26 Th wide (overlap 1 $m/z$) and a $1/k_0$ range of 0.3. Collision energy and resolution settings were the same as in the DDA method. Two windows were acquired per TIMS scan (100 ms) so that the total cycle time was 1.7 s.

## Library creation

Raw Bruker MS data were processed directly with Spectronaut 17.0.2 (Biognosys, Schlieren, Switzerland). A library was constructed from the DDA data for the fractions by searching the reference *A.thaliana* proteome (RefProt, www.UNIPROT.org) database of June 6, 2021 (39,335 sequences). For identification, peptides of 7–52 AA length were considered, cleaved with Trypsin/P specificity and a maximum of 2 missed cleavages. Carbamidomethyl-Cys (fixed), Met oxidation, and N-terminal protein acetylation (variable) were the modifications applied. Mass calibration was dynamic and based on a first database search. The Pulsar engine was used for peptide identification. Protein inference was performed with the IDPicker algorithm. PSM, peptide, and protein identifications were all filtered at 1% FDR against a decoy database.

Specific filtering for library construction filtered out fragments corresponding to less than 3 AA and fragments outside the 300–1800 $m/z$ range. Also, only fragments with a minimum base peak intensity of 5% were kept. Precursors with <3 fragments were also eliminated, and only the best 6 fragments were kept per precursor. For quantitation, only PSMs with a maximum of one missed cleavage were used. No filtering was done on the basis of charge state. Shared (non-proteotypic) peptides were kept.

The library created contained 100,634 precursors mapping to 75,018 peptide sequences, of which 47,963 were proteotypic. These corresponded to 9733 protein groups (12,446 proteins). Of these, 1548 were single hits (one PSM). In total 592,280 fragments were used for quantitation.

## DIA quantitation

Peptide-centric analysis of DIA data was done with Spectronaut 17.0.2 using the library described above. Both MS1 and MS2 data were used for quantitation[67]. Run alignment was assisted by a deep learning algorithm based on sample-specific in silico prediction of retention times of identified peptides and alignment based on local nonlinear regression. Interference correction (from neighboring isotope envelopes) was performed at both the MS1 and MS2 levels using windows of 2 and 3 min, respectively. Single hits proteins (defined as matched by one stripped sequence only) were kept in the Spectronaut analysis. Peptide quantitation was based on XIC area, for which a minimum of 1 and a maximum of 3 (the 3 best) precursors were considered for each peptide, from which the median value was selected. Peptides were retained for the calculation of protein group quantities if they passed the set $Q$-value threshold of identification (0.01) in at least 50% of the runs. For the retained peptides, any missing values were imputed with low-shifted values based on the global distribution of values in the entire experiment. Quantities for protein groups were obtained by summing all assigned peptide intensities after filtering. Global normalization of runs/samples was done based on the median of peptides.

In total, 96,237 precursors in the library were quantitated in the samples with the DIA data (cumulative on all runs). These corresponded to 73,259 peptides, resulting in 12,034 inferred proteins in 9448 protein groups. Mass tolerances were determined dynamically and were typically between 9 and 12 ppm. Average mass errors in individual samples were between 2.7 and 4 ppm in MS1 and MS2, respectively. The average of data points per peak was 7.07. Total normalized intensities for Protein Groups were exported to txt files for further analysis.

The mass spectrometry proteomics data have been deposited to the ProteomeXchange Consortium via the PRIDE partner repository (proteomexchange.org) with the dataset identifier PXD050814.

Data is accessible on the Project Webpage: http://www.ebi.ac.uk/pride/archive/projects/PXD050814. FTP Downloads are available at https://ftp.pride.ebi.ac.uk/pride/data/archive/2024/12/PXD050814.

## Data processing and statistical tests

All subsequent proteomics analysis was done with the Perseus software package[68]. Intensity values were log2-transformed. After the removal of contaminants, the table was filtered to retain only proteins quantitated in a minimum of 3 replicates per condition (7281 protein groups). Missing values were imputed by assigning them random values from a down-shifted normal distribution with standard Perseus parameters. After assignment to groups, a Student's $t$-tests were carried out among all conditions, with Benjamini–Hochberg correction for multiple testing ($q$-value

threshold > 0.05)[69]. The difference of means obtained from the test was used for 1D enrichment analysis on associated GO/KEGG annotation as described[70]. The enrichment analysis was also FDR-filtered (Benjamini–Hochberg, $q$-val < 0.02). Hierarchical clustering in two dimensions was performed with default Perseus parameters with Euclidean distance calculations, preprocessing by $k$-means, and using averages to calculate distances between clusters. Ten iterations were performed.

Peptide intensity values were normalized by dividing them by the total quantity of the parent protein in each sample. For the calculation of fold changes and statistical testing with peptide intensities, missing data were imputed only in the Col RL columns.

## Reporting summary
Further information on research design is available in the Nature Portfolio Reporting Summary linked to this article.

## Data availability
Uncropped Western blots and replicates corresponding to Figs. 2d, 8c are available in the Supplementary Information file. Source data for Figs. 2b, c, 3b, 8b are available in Supplementary Data 3. The mass spectrometry proteomics data have been deposited to the ProteomeXchange Consortium via the PRIDE partner repository (proteomexchange.org) with the dataset identifier PXD050814. Data is accessible on the Project Webpage: http://www.ebi.ac.uk/pride/archive/projects/PXD050814. FTP Downloads are available at https://ftp.pride.ebi.ac.uk/pride/data/archive/2024/12/PXD050814.

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

## Acknowledgements

Funding was received from the Swiss National Science Foundation (Grant 310030_208000) and the University of Neuchâtel for research on ABC1-like kinases.

## Author contributions

J.C. participated in the design of the experiment, carried out experiments and sample preparation, analyzed data, interpreted results, and participated in the writing of the original draft and review and editing. M.Q. performed proteomic experimentation, analyzed data, and participated in the writing of the original draft. R.P. and V.D. performed microscopy analysis. F.L. participated in the design of the experiment, analyzed data, interpreted results, and participated in the writing of the original draft. F.K. supervised the study, participated in the design of the experiment, analyzed data, interpreted results, and participated in the writing of the original draft and in review and editing.

## Competing interests

The authors declare no competing interests.
