## [Peer Review file · Communications Biology]

Arabidopsis conditional photosynthesis mutants *abc1k1* and *var2* accumulate partially processed thylakoid preproteins and are defective in chloroplast biogenesis

Corresponding Author: Professor Felix Kessler

This manuscript has been previously reviewed at another journal. This document only contains information relating to versions considered at Communications Biology.

Version 0:

Reviewer comments:

Reviewer #1

(Remarks to the Author)

I think the revisions strongly improved the manuscript and that it could be published now.

Minor: please add a label to the y-axis of the new Fig 3b.

Reviewer #2

(Remarks to the Author)

In this revised manuscript, Collombat et al. demonstrate that two unrelated mutants, *abc1k1* and *var2*, when subjected to overexcitation of PSII, fail to fully process a subset of their thylakoid proteins. As a result, the levels of the mature forms of these proteins are reduced and their processing intermediates accumulate. This is shown by two different analytical tools: immunoblot and MS analysis. The immunoblots clearly show the unprocessed bands and the MS analysis reveal the accumulation of peptides corresponding to the targeting sequences. These observations are interesting and novel, and although not fully understood mechanistically, I find them worth publishing.

A few minor points still need to be improved:

1. The abstract describes mostly background information and less emphasis is put on the results of the study. It should be rewritten.
2. l. 89-90 and elsewhere – To be more accurate, the *var2* mutant lacks FTSH2 but it still contains reduced levels of the thylakoid FTSH complex, because the other subunits, FTSH1, FTSH5 and FTSH8 are present. Total absence of the FTSH complex leads to lethality.
3. As explained in the Response to Reviewers, the small size of the seedlings did not allow the authors to do any work on isolated chloroplasts and thylakoids. As a result, we do not know where do the processing intermediates accumulate, in the stroma or the lumen. In other words, we cannot conclude whether the phenomenon is due to defects in translocation across the thylakoid membrane (or insertion into it) or in proteolytic processing in the lumen. This gap of knowledge should be made clear.
4. It appears that incomplete processing was observed mostly in abundant thylakoid proteins. Does this mean that the incomplete processing is not limited to a subset of thylakoid proteins? Please clarify this.

First, we would like to thank Reviewers 1 and 2 for their insightful comments. We believe they have helped us submit a much improved manuscript.

Reply to reviewers

The study by Felix Kessler and colleagues presents an intriguing comparison of mutants that exhibit defects in photosynthesis. The mutants were lacking the Ftsh protease (var2) or the pgr6 protein (abc1k1), respectively. Both lines exhibit impaired electron transport and display severe phenotypes, when photosystems are predominantly excited by red PSII light. The authors conducted a careful analysis of these defects covering morphological comparison, immunoblotting, and proteomics. The defect is most striking at the ultrastructural level, with an evident loss of the typical thylakoid membrane architecture observed in mutants grown for several days under red light (RL) conditions. Consequently, several proteins associated with chloroplast function and photosynthesis are present at lower levels in the mutants.

Interestingly, the authors observed the accumulation of specific precursor proteins targeted to the inner thylakoid lumen. For these proteins, certain peptides from the transit sequence were enriched in the proteomic dataset. The authors concluded that the specific photosynthetic defects of var2 and abc1k1 result in defective translocation via the Sec and TAT pathway. Such a malfunction could be attributed to a lack of energisation of the thylakoid membrane.

Firstly, the back-to-back comparison of the var2 and abc1k1 mutants is of considerable interest. It is notable that red-light exposure causes highly comparable defects in the two lines, although the severity of the defects differs. However, the significant disruption to chloroplast homeostasis also raises several concerns regarding the potential how specific mechanisms can be learned from the observed phenotypes. Despite the clear accumulation of certain precursor forms of the lumen proteins, it remains unclear whether this represents a specific defect for thylakoid lumen proteins or results from a more general problem of protein biogenesis and translocation.

To address the question whether this reflects a general problem of protein biogenesis and translocation, we have re-analyzed the proteomics data. The new results presented in Fig. 3b show that median log₂ fold changes for GO terms related to photosynthesis and specifically also those related to thylakoid lumen decrease more strongly than general chloroplast terms. This is an indication that it is not a general problem of biogenesis or translocation (see lines 314 to 321 in the discussion).

As mentioned by the authors, the accumulating precursor proteins are highly abundant. It is thus possible that other non-lumen precursor proteins are similarly accumulating but that these proteins were not detected.

To address whether other non-lumen precursor proteins similarly accumulate (i.e. targeting peptides increase under red light/mature peptides decrease) we have done three things: 1) we have added a new Fig. 5a. Fig. 5a shows a heat map of all 267 peptides measured for Photosystem II (PsbX), Photosystem I (PsaX) and the cytochrome b6f (PetX) components. Being components of the electron transport chain, their abundance is similar to the

accumulating precursor proteins. But except for the peptides already identified in the manuscript none were upregulated under red light.

2) In new Table 3, we show the analysis of all chloroplast proteins. In addition to the already identified components we have identified 14 additional preproteins that accumulated similarly but in a less pronounced way (lines 269-272 and 335-339). Most of these are thylakoid proteins (seven related to LHCB, two additional thylakoid lumen proteins). This is less than 1% of all chloroplast proteins (GOCC). We conclude that relatively few preproteins accumulated similarly to the initially identified preproteins.

3) (same as one of the replies to comment on Figure 5) New Fig. 6a looks at 71 thylakoid lumen proteins and show that the phenomenon is specific for the identified set of luminal proteins except for two proteins (Table 3) that showed the phenomenon to a lesser extent (lines 269-272 and 346-350).

One can hypothesize that the severe CP defect may also result in the accumulation of all imported proteins, and that the chloroplast defect generally impairs translocation.

*To address the point whether the severe CP defect results in the accumulation of all imported proteins, we have conducted a more detailed inspection of the proteomics data. As shown in new Fig. 3b, GO terms linked to photosynthesis-associated proteins (photosystems and O₂ evolving complex) were significantly more depleted than the average of all chloroplast proteins. The volcano plot in Fig. 3C illustrates that many chloroplast proteins were even upregulated in *abc1k1* compared to the *Col-0* wildtype.*

I am worried that the 5 day exposure to RL was too long. It would be of interest to ascertain whether a short-term exposure to RL would promptly affect protein translocation. In such a scenario, there would be a more direct correlation between photosynthesis and thylakoid translocation.

We observed that the accumulation of partially processed preproteins is comparatively slow and does not become visible immediately after a short-term exposure to RL. It takes several days for them to accumulate and become detectable.

However, is it really a novel observation that thylakoid translocation is reduced when photosynthesis is impaired and thus ATP levels and a proton gradient are reduced?

We have scoured the literature and, rather astonishingly, were unable to find any reports referring to the requirement of photosynthetic activity for preprotein processing and for chloroplast biogenesis. The use of strong but conditional mutants has facilitated our research and allowed to address this (in our minds) important question.

The authors need to better elaborate what was specifically learned by using these mutants. This is not clear throughout the results and discussion in the current manuscripts.

We try to be clearer. From the new analyses, we have learned that the processing and accumulation of essential thylakoid lumen and several additional thylakoid proteins (chlorophyll-binding proteins and ATP synthase subunits) are strongly impaired while around 99% of chloroplast proteins are moderately affected or even increased in the conditional

photosynthesis mutants (Figures 3b and 3c). Based on the sum of observations, “we hypothesize that the onset of photosynthesis and processing of the observed set of thylakoid preproteins are mutually dependent constituting a critical step in chloroplast biogenesis” (lines 389-391).

In case the mutants result in a highly specific defect of thylakoid membrane processes, while leaving other CP processes and other cellular defects unaltered, this would be certainly of interest. However, in that case, it would be important to conduct a more thorough analysis of the proteomic data and to clearly distinguish between the specific defects and the processes that are not altered.

*New Figure 3b shows median log₂ fold changes for GO terms related to photosynthesis (containing many thylakoid proteins) and general chloroplast. Comparison shows that GO terms related to photosynthesis decrease far more (around 50%) than terms related to the chloroplast in general (around 10%) (lines 314-321. This can be taken as an indication that CP processes apart from photosynthesis are not likely to be strongly altered. We also noticed that the GO-term related to peroxisome (comprising 145 proteins) was upregulated in the *abc1k1* mutant compared to the *Col-0* wildtype. This suggests that energy production using seed reserves remains functional despite the photosynthesis defect (lines 376-378).*

Prominent processes that are not affected should be also shown in the figures. Judging from the volcano plot, I would rather assume that the mutants exhibit significant, globular changes when exposed to red light for five days.

We have now specifically compared photosynthesis related to general chloroplast, peroxisome, proteasome and ribosome GO terms (Fig. 3b). The median fold changes (log₂) showed a much more pronounced reduction for the photosynthesis terms than for general chloroplast terms. Peroxisome and proteasome terms were even slightly upregulated. Even though, the mutants exhibit significant global change under red light, the amplitude is not as large as one might assume.

Minor points:

Figure 3: Please indicate the number of proteins that are generally altered in the proteome of the mutants.

*These numbers are given in Table 1 and on lines 201 to 203 for *abc1k1*: All RL samples showed strong differences when compared to their WL controls, with almost one third of the proteome impacted by RL in the *abc1k1* genotype. 974 proteins were altered in *var2* RL and 965 in *abc1k1* RL in both cases equalling around 10% of total when compared to *Col0* RL.*

It would be of interest to ascertain whether these proteins are predominantly located in the chloroplast.

*There are global changes of varying magnitude in all cellular compartments. Again, in the comparison *ABC1k1* RL versus *Col* RL, the photosynthetic machinery concentrates by far the strongest changes in a defined compartment, as shown by the GO term enrichment analysis.*

The wider chloroplast on average shows only a mild, though pervasive, decrease (log2 FC - 0.1 to -0.2). Interestingly, there are also chloroplast proteins that are upregulated (Fig.3C). We think that we have better explained these aspects in the revised version of the manuscript.

It needs to be clarified, whether the observed difference is log2-based, as assumed by “mutant minus Col”.

Yes, it is log2-based

Please clarify whether a cut-off was applied for the enrichment.

A fold change of 1 was applied as cut-off for the enrichment.

It would be good to include protein names and/or categories (e.g., psax, psbx) alongside the orange dots.

We have now modified the volcano plot in Fig. 3c. The green dots all represent PsaX, PsbX and PetX proteins (the ones shown in Fig. 4a). But we chose not to include the names as this would make the figure illegible. The blue dots indicate chloroplast proteins. It is interesting to note the existence of a surprisingly large number of upregulated chloroplast proteins in abc1k1 compared to Col-0.

It would be beneficial to present the enriched GO classes in a main figure.

We show GOCC GO terms enriched for photosynthesis and general chloroplast terms also including the terms ribosome, proteasome complex and peroxisome in new Fig. 3b. Table 2 shows medians from 1D enrichment of selected KEGG categories Photosynthesis, Ribosome, Proteasome, Peroxisome). For the complete list please consult Supplementary_Table_1

Figure 4 and the paragraph commencing at line 213...

The relationship between Col-RL and the WL samples is not particularly unexpected, given that it was already observed in the PCA. The shown cluster is heavily dominated by chloroplast-encoded proteins, which may not be subject to regulation if the light regime is changed from WL to RL. In the mutant, there appears to be a disturbance in the overall CP protein homeostasis.

Figure 4a indicates that except for PsbW all proteins in the cluster are subject to downregulation under red light (comparison of Col RL to Col WL or ABC1k1 WL to Abc1k1 RL columns). Moreover, the new analysis (Fig. 3b) shows that general chloroplast terms are far less affected compared to photosynthesis-related terms) pointing to a specific photosynthesis defect (lines 378-380).

Figure 5. It would be important to present the intensities of other major thylakoid proteins (e.g., LHCs) to demonstrate the specificity of this phenomenon.

I was unable to evaluate the Supplemental Table, as it was not provided.

We address this point in the 3 ways:

- 1) Fig. 5b represents all proteins related to the Photosystems and the cytochrome b6f complex (PsaX, PsbX and PetX proteins). Their abundances are stoichiometrically comparable to the partially processed preprotein of the thylakoid lumen but they do not show the phenomenon.*
- 2) Furthermore, new Fig. 6a looks at 71 thylakoid lumen proteins and show that the phenomenon is specific for the identified set of luminal proteins except for two proteins (Table 3) that showed the phenomenon to a lesser extent (lines 269-272 and 346-350).*
- 3) Further analyses of all chloroplast proteins (Table 3) identified additional thylakoid proteins (seven LHC-related proteins and two subunits of the ATP synthase) and these were also affected to a lesser extent. We mention this in the text and have modified it to take these findings into account (lines 269-79 and 335-339).*

Sorry, we have now provided the Supplemental Tables

Reviewer #2 (Remarks to the Author):

Aiming to determine “whether photosynthesis itself is required for chloroplast biogenesis”, Collombat et al. have nicely performed a set of experiments on 5-days old seedlings of WT, *abc1k1* and *var2* mutant genotypes, germinated and grown under white light (WL) or red light (RL). These seedlings were subjected to various analyses, including transmission electron microscopy, immunoblotting, photosynthesis measurements and a detailed proteomic analysis. The authors show that under WL all genotypes develop, more or less, similar to WT, their chlorophyll content is close to normal and so is their PSII quantum yield, and the ultrastructure of their chloroplasts (Figs. 1 and 2). In contrast, under RL the mutant seedlings are smaller than WT and yellowish, they contain less chlorophyll and their PSII quantum yield is reduced. Moreover, they do not contain typical thylakoids in their plastids. Consistent with these, the level of a number of photosynthetic proteins is reduced under RL, and interestingly, some luminal proteins appear to accumulate in their unprocessed form (Fig. 2d). The proteomic analysis highlights the reduction in the level of many photosynthetic proteins in the mutant genotypes grown under RL (Figs. 3 and 4), and demonstrate that 12 PSI and PSII proteins are not correctly processed to their mature forms and still contain parts of their N-terminal targeting sequences (Figs. 5-7 and Table 2). Lastly, the phenotype of the *abc1k1* mutant, but not of *var2*, can be partially repaired by adding to the growth plates a low concentration of the PSII inhibitor DCMU (Fig. 8). Taken together, the authors claim that these results support the notion that normal chloroplast development requires active photosynthesis already at the earliest stages of seedling development.

From a technical point of view, I find this work flawless.

Thank you, we appreciate this comment!

The methods, the experiments and their results are all clearly explained and presented. However, reading the manuscript raised some questions and thoughts in my mind, as follows:

1. The accumulation of unprocessed forms of some photosynthetic proteins in the mutants under RL, observed both in immunoblots and in the proteomic analysis, is interesting. However, the authors interpret this observation as an indication for failure to correctly process in response to the impaired photosynthesis in the mutants. As all these proteins are located in the lumen, they need to be first translocated across the thylakoid membrane before they are processed. In the absence of thylakoids (Fig. 1C), it is similarly likely that these proteins fail to translocate across a thylakoid or pre-thylakoid membrane. Without analyses at the level of isolated chloroplasts and isolated thylakoids, one cannot conclude whether we see here impaired translocation into the lumen or impaired processing in it.

This is a pertinent remark. However, it is extremely difficult to address this experimentally as the plants used in the study are so tiny that we were unable to isolate chloroplasts or thylakoids. However, apart from the 7 proteins already identified and two additional proteins (new Table 3) a list of 71 thylakoid lumen proteins appeared to be normally processed. But, we mention in the discussion (lines 361-363) that we were unable to conclude whether its impaired translocation or impaired processing).

2. A number of steps during the biogenesis of chloroplasts require ATP (e.g., protein import across the envelope, protein translocation across the thylakoid membrane, etc.). As many of these proteins are components of the photosynthetic machinery, their translocation and targeting in early stages of biogenesis are likely to rely on ATP generated in mitochondria and imported into plastids before the photosynthetic machinery starts to assemble and generate ATP within the organelle. (To the best of my knowledge, this issue has never been approached experimentally). Thus, I find the argument that chloroplast biogenesis requires active photosynthesis somewhat shaky. How early is early in chloroplast biogenesis?

We have changed “early in chloroplast biogenesis” to “the onset of seed germination”(lines 118 and 136). At this stage it is possible to capture the initial accumulation of photosynthesis-associated proteins at the beginning of chloroplast biogenesis. Regarding energy provision from seed stores, new Fig. 3b shows that the GO term peroxisome is upregulated in abc1k1 compared to the Col-0 wildtype. This suggest that energy provided from seed stores may be used to compensate in part for the loss of photosynthesis in the mutants. But, based on the data, it does not appear to compensate for thylakoid protein transport.

3. I find studying chloroplast biogenesis under RL somewhat an artificial situation. Doing that in mutant backgrounds complicates the issue even more. Thus, drawing conclusions about sequential events in chloroplast biogenesis, in this case – establishing photosynthetic capacity before translocation or processing of luminal proteins can occur, is difficult if not impossible, at least with the experimental approach presented here.

We try to be cautious regarding “sequential events”. On lines 384 to 385 we mention the “the association of processing with photosynthesis...” rather than establishing a causal link. On lines 389 to 391 “we hypothesize that the onset of photosynthesis and processing of the

observed set of thylakoid preproteins are mutually dependent constituting a critical step in chloroplast biogenesis”.

4. ABC1K1 is supposedly a protein kinase whose biochemical and molecular functions within chloroplasts are far from being well understood. For instance, is the kinase activity of ABC1K1 necessary for the WT phenotype? Will a variant lacking kinase activity complement the mutant phenotype of *abc1k1*?

*This is an interesting question that we have addressed in a separate manuscript showing that a variant mutated at the active site aspartate 400 does not complement the *abc1k1* phenotype. A revised manuscript describing these results has been resubmitted. Moreover, a mutagenesis study of a yeast homolog suggests that ATPase activity is required for function. We have now added information on lines 99-101.*

5. VAR2, one component of the thylakoid FTSH protease, is better characterized with respect to PSII repair. However, the possible role of both FTSH and ABC1K1 in chloroplast biogenesis is not known. This work does not advance our knowledge on this issue either.

*We do not postulate a direct role for either FTSH or ABC1K1 in chloroplast biogenesis. We rather use the *var2* and *abc1k1* mutants as tools to conditionally disrupt photosynthesis under red light which in turn leads to failure of chloroplast biogenesis. We hence propose that the role of both proteins in chloroplast biogenesis is to assure the onset of functional photosynthesis and is therefore indirect in nature.*

Minor points:

1. In the Introduction, more information about the biochemical and molecular functions of ABC1K1 could be useful.

In the revised version we have included additional information on the biochemical and molecular function of ABC1K1 (lines 97 to 105)

2. p. 4, Fig. 1 – What does Fig. 1b show? You do not refer to it in the text.

Fig. 1b shows germinating seedlings of the different genotypes at 48h and 72h under white and red light respectively. We have now included a reference in the text (line 141)

3. l. 326 – What do you mean by “red-light switch”?

We removed this formulation and now just say “switch off photosynthesis using RL”. Line 309.

4. Font size in many of the annotations in the figures is too small.

We have improved this in some of the figures and will complete it in the final accepted version of the manuscript.

In summary, although the results presented here are interesting, I find that this work does not advance our understanding of neither chloroplast biogenesis at large, nor of any specific step within this multi-faceted process.

Reply to Reviewer's comment :

We thank both Reviewers for taking their time to review the revised version of our manuscript.

Reviewer 1:

Thank you for your approval of our revised manuscript.

Reviewer 2:

Thank you for your comments. I address them point by point in the following:

1. The abstract describes mostly background information and less emphasis is put on the results of the study. It should be rewritten.

We have removed some of the background information and put more emphasis on the results of the study. Please check the marked-up version of the manuscript.

2. l. 89-90 and elsewhere – To be more accurate, the var2 mutant lacks FTSH2 but it still contains reduced levels of the thylakoid FTSH complex, because the other subunits, FTSH1, FTSH5 and FTSH8 are present. Total absence of the FTSH complex leads to lethality.

Lines 90 to 92: we now say “However, reduced levels of the thylakoid FTSH complex remain as the other subunits FTSH1, FTSH5 and FTSH8 are still present and prevent the lethal phenotype observed in the complete absence of the FTSH complex.”

3. As explained in the Response to Reviewers, the small size of the seedlings did not allow the authors to do any work on isolated chloroplasts and thylakoids. As a result, we do not know where do the processing intermediates accumulate, in the stroma or the lumen. In other words, we cannot conclude whether the phenomenon is due to defects in translocation across the thylakoid membrane (or insertion into it) or in proteolytic processing in the lumen. This gap of knowledge should be made clear.

Lines 362-366: we say “However, from the experiments in the present study, limited by the tiny size of the seedlings, it cannot be concluded whether the partially processed preproteins resulted from non-translocation or non-

processing by PLSP1. This represents a gap of knowledge that we plan to address in the future.”

4. It appears that incomplete processing was observed mostly in abundant thylakoid proteins. Does this mean that the incomplete processing is not limited to a subset of thylakoid proteins? Please clarify this.

Lines 346 to 350: we attempt to clarify this point by saying ”The subset of seven partially processed pre-proteins of the oxygen evolving complex and the plastocyanin docking site are localized in the thylakoid lumen and are part of the thylakoid lumen proteome of 71 proteins. Amongst the 71 thylakoid lumen proteins only two others, less abundant, showed comparable but less pronounced behaviour whereas no other thylakoid lumen proteins produced peptides derived from targeting sequences.” Also, amongst many stromal proteins only 3 produced peptides derived from targeting sequences (lines 335 to 336). The phenomenon is thus quite clearly limited to a subset of thylakoid proteins.